# Characterization of surface clutter signal in presence of orography for a spaceborne conically scanning W-band Doppler radar

Francesco Manconi[1], Alessandro Battaglia[1,2], and Pavlos Kollias[3,4]

[1]DIATI, Politecnico of Torino, Turin, Italy
[2]Department of Physics and Astronomy, University of Leicester, Leicester, UK
[3]School of Marine and Atmospheric Sciences, Stony Brook University, Stony Brook, NY, USA
[4]Department of Atmospheric and Oceanic Sciences, McGill University, Montreal, QC Canada

**Correspondence:** francesco.manconi@polito.it

**Abstract.** The Earth's surface radar reflection is one of the most important signals received by spaceborne radar systems. It is used in several scientific applications including geolocation, terrain classification, and path-integrated attenuation estimation. A simulator based on the ray tracing approach has been developed to reproduce the clutter reflectivity and the Doppler velocity signal for a conically scanning spaceborne Doppler radar system. The simulator exploits topographic information through a raster Digital Elevation Model, land types from a regional classification database, and a normalized radar surface cross-section look-up table. The simulator is applied to the WInd VElocity Radar Nephoscop (WIVERN) mission, which proposes a conically scanning W-band Doppler radar to study in-cloud winds. Using an orbital model, detailed simulations for conical scans over the Piedmont region of Italy that offers a variety of landscape conditions are presented. The results highlight the strong departure of the reflectivity and Doppler velocity profiles in the presence of marked orography and the significant gradient in the surface radar backscattering properties. The simulations demonstrate the limitations and advantages of using the surface Doppler velocity over land as an antenna-pointing characterization technique. The simulations represent the full strength range of the surface radar clutter over land surfaces for the WIVERN radar. The surface clutter tool applies to other spaceborne radar missions such as the nadir pointing EarthCARE and CloudSat cloud profiling radars, or the cross-track scanning GPM precipitation radars.

## 1 Introduction

Space-borne atmospheric radars in bands between X and G (i.e. from 10 to 300 GHz) are now considered cornerstones of the global observing system for characterising vertical profiles of clouds and precipitation systems (Battaglia et al., 2020). While Ku-Ka and W bands have been used in space for more than a decade (Kummerow et al., 1998; Skofronick-Jackson et al., 2016; Stephens et al., 2018; Illingworth et al., 2015), new frequency bands are currently being explored (Battaglia et al., 2014; Li et al., 2020), with novel scanning modes (e.g. conical scanning as proposed in Illingworth et al. (2018)) and innovative Doppler capabilities (Battaglia et al., 2013; Tanelli et al., 2016; Kollias et al., 2022). Compared to ground-based radars, space-based radars provide a global perspective and are particularly well suited to studying clouds in the upper troposphere, where attenuation by water vapour and liquid-phase hydrometeor is less pronounced. Conversely, space-based observations are hampered

by the strong surface return (hereafter referred to as "clutter") that tends to obscure the hydrometeor signal near the ground. Knowing the shape of the clutter reflectivity allows the signal to clutter ratio to be determined. This parameter is an indication of the "blind zone" near the surface and is crucial for the correct quantification of surface precipitation (Maahn et al., 2014), the detection of shallow clouds (Burns et al. (2016); Lamer et al. (2020)) and the measurement of near-surface winds. Meneghini and Kozu (1990) suggested that the blind zone can be significantly reduced when scanning at high angles of incidence (similar to scatterometers) due to the reduced surface normalised radar cross section (NRCS) when moving away from the nadir looking configuration.

The WIVERN mission, short for WInd VElocity Radar Nephoscope, www.wivern.polito.it (Illingworth et al., 2018; Battaglia et al., 2022a; ESA-WIVERN-Team, 2023), one of the two remaining candidates in ESA's Earth Explorer 11 programme, proposes a W-band conically scanning radar with an angle of incidence of about 42°. Usually, for atmospheric radars, the surface is considered a disturbance; an important matter is to assess how large this disturbance is, i.e. to quantify the signal-to-clutter ratio. It is therefore timely to investigate and assess how beneficial such a scanning configuration could be in terms of reducing/enhancing the signal-to-clutter ratio for all type of surfaces (and not only for "flat" oceanic).

On the other hand, the presence of the surface return represents an opportunity because it provides a reference point that can be used either to derive the path integrated attenuation via the surface reference technique (Meneghini et al., 2000), to calibrate the reflectivity (Tanelli et al., 2008) and/or the Doppler velocity (Battaglia and Kollias, 2014; Scarsi et al., 2024) and/or to provide accurate geolocation (Puigdomènech Treserras and Kollias, 2024). In particular, based on simulations for flat homogeneous surfaces for the WIVERN radar specifics, Scarsi et al. (2024) showed that the clutter Doppler velocity profiles (expected to be 0 m/s in correspondence to the surface reflectivity peak) can be used for mispointing corrections. The mission requirements for the horizontal component of the line-of-sight wind measurements is in the order of 2.5 m/s: to achieve this goal, the contribution on mispointing errors must be lower than 0.4 m/s after all possible calibration methods (ESA-WIVERN-Team, 2023). The on-board attitude determination and control system can provide pointing and knowledge of it within a certain degree of accuracy, which may not be sufficient to satisfy the scientific requirements. Thermoelastic deformations of the antenna will also largely contribute to the pointing error as demonstrated by the recently launched EarthCARE Doppler radar (Kollias et al., 2023). This effect is cyclical with the orbital period but it is hard to model and predict via numerical models driven by the antenna properties (e.g. by its temperature at different locations). These effects can be corrected by external calibration methods (Scarsi et al., 2024), with the surface being the simplest natural target. However, in presence of real land surfaces, clutter Doppler velocity and reflectivity profiles are expected to deviate significantly from the profiles obtained for homogeneous flat surfaces for two reasons:

1. the variability of surface height within the radar footprint introduced by the orography, which will alter the iso-range lines;

2. the inhomogeneity of the surface backscatter cross section within the radar footprint (the so called non uniform beam filling, NUBF, Tanelli et al. (2002)), that will bias the Doppler velocity signal towards the velocities of the brightest regions.

Better understanding the shape of the clutter reflectivity and of the clutter mean Doppler velocity profiles is paramount for two reasons: 1) the reflectivity profile can be used for geolocation purposes (Puigdomènech Treserras and Kollias, 2024) and its shape is relevant for assessing the blind zone of a radar system (i.e. the region where the radar signal will not provide any useful information for the hydrometeors); 2) the surface Doppler can be used in a data-driven approach to mitigate mispointing errors. Since the calibration of antenna distortions can occur on shorter time scales more frequent calibration points are needed. Therefore, it must be assessed what surfaces can be useful for this kind of purpose by quantifying the limits of acceptable variability in terms of sigma-zero and orography. Therefore, the aim of this work is to extend the simulations of the clutter signal to non-planar surfaces (characterised by a very high resolution DEM), including a realistic variability of the surface backscatter (based on a surface classification index). A geometric-optical approach is used, similar to that used in Delrieu et al. (1995); Gabella and Perona (1998); Gabella et al. (2008) for ground-based weather radars. The novelty is the application to a space-based configuration, the extension to the Doppler signal and the inclusion of NUBF effects. The simulator will be applied to several case studies and an initial assessment will be made of how much the shape of the reflectivity and Doppler velocity profiles are distorted from those expected for a flat homogeneous surface.

This clutter simulator represents a module of a larger end-to-end simulator endeavour being developed as part of the phase A activity funded by ESA, which simulates the full return from both atmospheric and surface targets. The whole simulator is based on the work already developed at Politecnico di Torino in the past five years (Battaglia et al., 2022b; Rizik et al., 2023; Battaglia et al., 2024). This work completes the simulator adding a thorough treatment of the surface accounting for variability of $\sigma_0$ at fine scales and orographic effects. In the previous simulator, the surface was treated in a simplistic way (flat and homogeneous) which is sufficient for oceanic surfaces. The simulator discussed in the present work will be integrated into the existing one for detailed studies that require a complete surface characterization. To the best of the authors' knowledge, there is currently no clutter simulator of reflectivity and Doppler signal for spaceborne radars taking into account NUBF for ground return and orography that has been developed for past (GPM, CloudSat, EarthCARE) or future missions (e.g. INCUS). Therefore, this type of work is a first, and it is very relevant for the WIVERN radar, given its Doppler capabilities and conical scanning operation.

After introducing the methodology (Sect. 2), examples of the simulation are illustrated in Sect. 3 for an overpass over the mountainous Piedmont region (northwest part of Italy). Finally a statistical analysis is presented in Sect. 4. Conclusions and future work are outlined in Sect. 5.

## 2 Methodology

The flowchart of the procedure that computes the surface clutter signal (reflectivity and Doppler velocity) is presented in Fig. 1. The software inputs are: 1) a raster Digital Elevation Model (DEM) map; 2) a surface class map; 3) the satellite orbit with the associated antenna scanning; 4) the antenna gain pattern; 5) a noise and receiver model; 6) a NRCS model for each surface class, based on LUTs derived from literature. These inputs are used to compute the clutter reflectivity and Doppler velocity profiles.

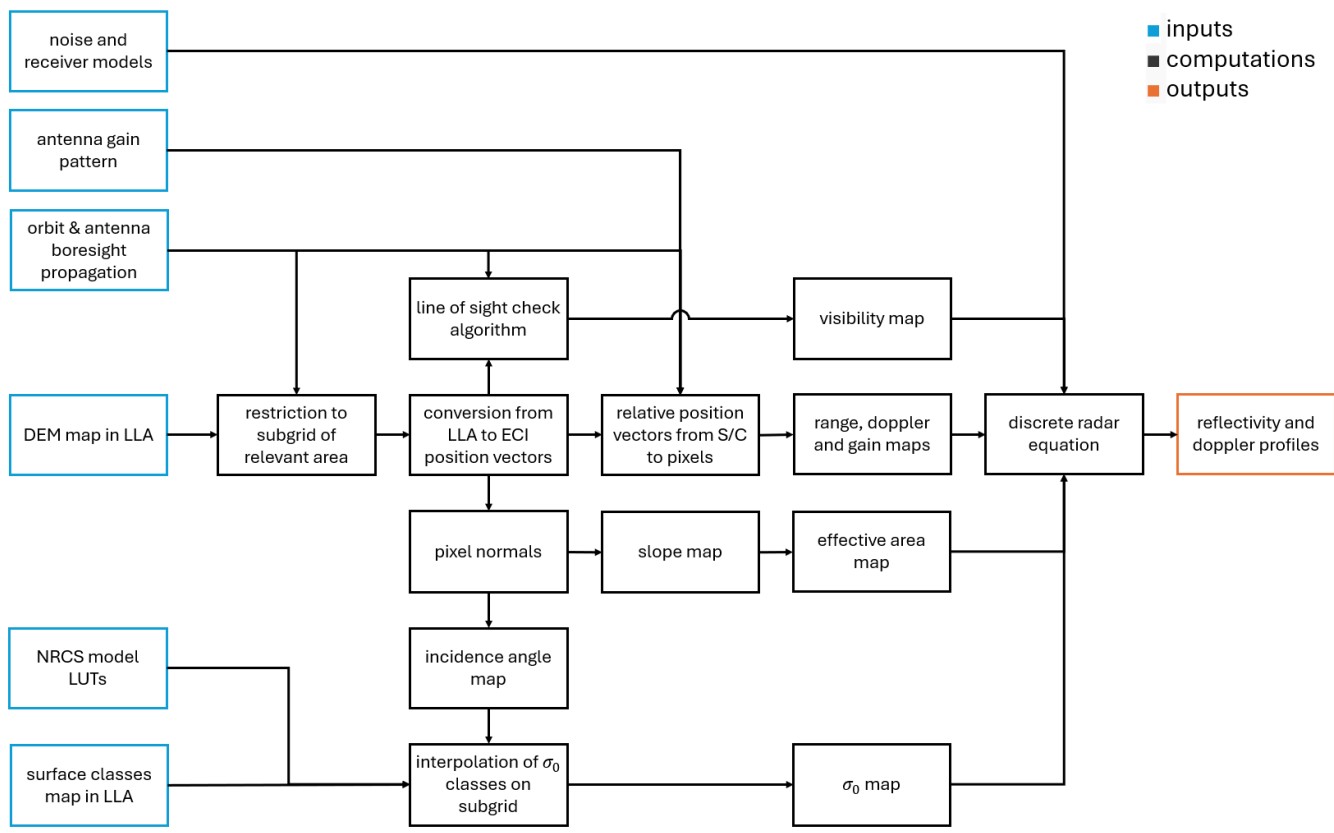

**Figure 1.** Flowchart of ground clutter computation code flowchart. LLA= latitude, longitude and altitude; ECI=Earth centered inertial.

## 2.1 Surface echo return: reflectivity and Doppler velocity

Fig. 2 shows the geometry for slant angle radar observations, where a pulse hits the surface at an angle of incidence, $\theta_{inc}$. It is assumed that the transmitted radar pulse has a top hat shape with a duration of $\tau_p$ and is transmitted according to an antenna pattern characterised by a main lobe (blue shaded cone) and various side lobes (black envelopes). Iso-range lines are shown on the illuminated orographic surface, one for the leading edge and one for the trailing edge of the top-hat pulse shape around the range of the intersection between the boresight axis and the DEM surface. WIVERN orbit and radar specifics are listed in Tab. 1. The radar will transmit and receive both H and V signals in pairs. Each pair has a repetition frequency of 4 KHz, with a delay between the H and V pulses of 20 $\mu$s (see Fig. 1 in Rizik et al. (2023) for a schematic of the concept).

The power received at any time $t$ (and the corresponding range $r = ct/2$, where $c$ is the speed of light) results from the contributions of targets located within the spheres centred on the radar and produced by the propagation of the trailing and leading edges of the pulse, shown in Fig. 2 as orange and green curves respectively. In the case of a flat surface, these targets include an annular strip of terrain (Battaglia et al. (2017)), but in the case of complex terrain such regions, identified in the

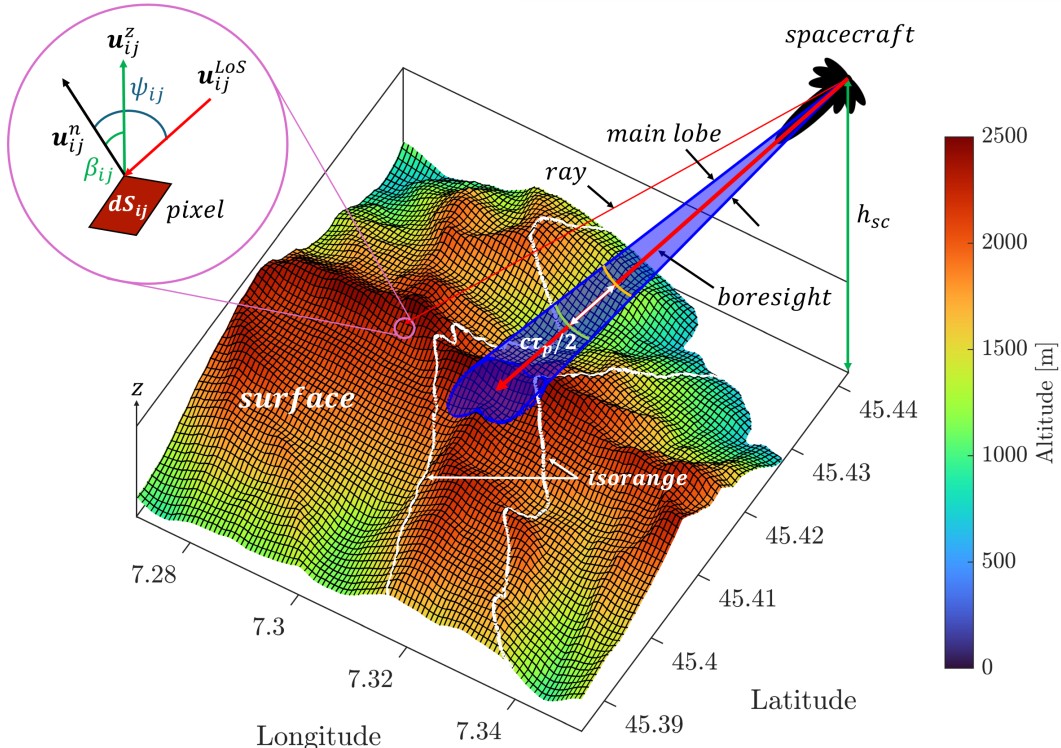

**Figure 2.** Geometry for a slant-looking radar illuminating a region with pronounced orography described by a high resolution DEM. The reflectivity and Doppler velocity at any given range $r$ are computed via an integral extended to the region comprises between $r - c\tau_p/4$ and $r + c\tau_p/4$ as described in Sect. 2.1 (Eqs. 4-6). Such surface region is divided in infinitesimal elements (black squares). All the relevant mathematical quantities are illustrated in the top left inset. The two white iso-ranges in the figure correspond to $r_{bs} - c\tau_p/4$ and $r_{bs} + c\tau_p/4$, where $r_{bs}$ is the range in the boresight direction.

following as $S$, become much more complicated and dependent on the illumination geometry and the orography. The power received by the radar from the surface at range $r$, $P_r$, assuming that the antenna gain is identical for transmission and reception,
is given by an integration performed over the illuminated area, $S$ (Meneghini and Kozu, 1990):

$$P_r(r) = \underbrace{\left[ P_t \frac{\lambda^2}{(4\pi)^3} G_0^2 \right]}_{C_S} \int_S \frac{\sigma_0(\psi) \, G_n^2 \, |u(2r/c - 2\xi/c)|^2}{r^4} \, dS \approx C_S \sum_{i,j} \frac{\sigma_0(\psi_{ij}) \, G_n^2(\mathbf{u}_{ij}^{LoS}) \, |u(2r/c - 2r_{ij}/c)|^2}{r_{ij}^4} dS_{ij} \tag{1}$$

where $P_t$ is the transmitted power, $\lambda$ is the wavelength of radar, $G = G_0 G_n$ is the antenna gain ($G_0$ being the maximum gain at antenna boresight), $u(t)$ is the complex voltage envelope of the transmitted pulse (for a top hat shape $|u(t)| = 1$ for $-\tau_p/2 < t < \tau_p/2$), $\xi$ is the distance between the infinitesimal element $dS$ and the radar, $\psi$ is the local incidence angle that
can be computed as $\psi = acos(\mathbf{u}_{ij}^{LoS} \cdot \mathbf{u}_{ij}^n)$ where $\mathbf{u}_{ij}^n$ is the normal to the infinitesimal surface. The integral (summation) is extended only to those pixels that are visible from the radar (see Sect. 2.1.1). The normalised radar cross section (NRCS), $\sigma_0$,

**Table 1.** WIVERN orbit and radar specifics, as currently under study in the Phase-A study for the ESA Earth Explorer 11 program.

| | |
|---|---|
| Spacecraft height, $h_{SC}$ | 500 km |
| Spacecraft velocity, $v_{SC}$ | 7600 ms$^{-1}$ |
| Orbit inclination, $i$ | 97.42° |
| Orbit Local Time of the ascending node, $LTAN$ | 06:00 |
| Incidence angle, $\theta_{inc}$ | 41.6° |
| Swath width at ground | 800 km |
| Radar output frequency | 94.05 GHz |
| Pulse width $\tau$ | 3.3 $\mu$s |
| Antenna angular velocity, $\Omega_a$ | 12 rpm |
| Antenna elevation beamwidth, $\theta_{3dB}$ | 0.0656° |
| Antenna azimuth beamwidth, $\phi_{3dB}$ | 0.0722° |
| Footprint speed | 500 kms$^{-1}$ |
| Single pulse minimum detectable reflectivity | -18 dBZ |
| H-V Pair Repetition Frequency | 4 kHz |
| Time between H and V pulses, $T_{HV}$ | 20 $\mu$s |
| Range sampling distance (rate) | 100 m (1.5 MHz) |
| Number of H-V Pairs per 1 km integration length | 8 |

is defined as the surface radar backscatter cross section, $\sigma_{surf}^{back}$ normalised to the surface area, $A$, and is typically expressed in $dB$ units as:

$$\sigma_0[dB] \equiv 10\log_{10}\frac{\sigma_{surf}^{back}}{A}. \qquad (2)$$

No attenuation effect has been included in the clutter simulation (it is included, however, in the full simulator). The Doppler signal is not affected by attenuation as far as the SNR remains high, as is the case for surface targets, and the shape of the reflectivity profile is also unchanged (atmospheric attenuation simply lowers the profile by the path integrated attenuation of the whole atmospheric column). On the right hand side of Eq. (1) the integral has been replaced by a summation over different small surfaces. $\mathbf{u}_{ij}^{LoS}$ is the line-of-sight unit vector joining the satellite to the surface element. Note that a radar constant $C_S$

relevant for a surface target has been introduced (square bracket in Eq. 1).

     The surface, locally at each pixel, is approximated with inclined plane facets of area $dS_{ij}$ whose inclination is given by $\beta_{ij}$ which is computed according to the local slope of the terrain. The area $dS_{ij}$ can be computed as a function of the DEM pixel area $\Delta x_i \, \Delta y_j$ by:

$$dS_{ij} = \frac{\Delta x_i \, \Delta y_j}{\cos(\beta_{ij})}$$

where $\beta_{ij}$ is the slope of the $ij$ surface element that can be derived as $\beta = acos(\mathbf{u}_{ij}^z \cdot \mathbf{u}_{ij}^n)$ where $\mathbf{u}_{ij}^z$ is the unit vector along the local vertical direction.

In radar meteorology for meteorological distributed targets the radar reflectivity is defined as:

$$P_r(r) = C_M \frac{Z}{r^2} \qquad \text{where } C_M \equiv \frac{\pi^2}{2^6} \frac{P_t\, G_0^2 \Omega_{2A}}{\lambda^2} \frac{c\tau_p}{2} |K_w|^2 = C_S \frac{\pi^5 |K_w|^2 \Omega_{2A}}{\lambda^4} \frac{c\tau_p}{2} \tag{3}$$

where $K_w$ is derived from the refractive index of water at 3 mm-wavelengths ($|K_w|^2$ assumed equal to 0.78), $\Omega_{2A} \equiv \int G_n^2 d\Omega$ (which for a Gaussian beam is approximately equal to $\frac{\pi \theta_{3dB} \phi_{3dB}}{8 \log(2)}$) and $C_S$ is another radar constant, previously defined in Eq. (1). Eq. (3) allows to convert $P_r$ to $Z_e$ for any given range as:

$$Z(r) = r^2 \frac{2\lambda^4}{\pi^5 |K_w|^2 \Omega_{2A} c\tau_p} \sum_{i,j} \frac{\sigma_0(\psi_{ij})\, G_n^2(\mathbf{u}_{ij}^{LoS})\, |u(2r/c - 2r_{ij}/c)|^2}{r_{ij}^4} dS_{ij}. \tag{4}$$

Note that for flat surfaces with constant NRCS, $\sigma_0$:

$$\int Z(r)dr = \frac{\lambda^4 \sigma_0}{\pi^5 |K_w|^2 \cos\theta_{inc}} \tag{5}$$

which provides a useful check for the normalization of the reflectivity profile.

The Doppler velocity at range $r$ is computed similarly to Eq. (4) as:

$$v_D(r) = \frac{C_S}{P_r(r)} \sum_{i,j} \frac{v_{SC}(ij)\sigma_0(\psi_{ij})\, G_n^2(\mathbf{u}_{ij}^{LoS})\, |u(2r/c - 2r_{ij}/c)|^2}{r_{ij}^4}\, dS_{ij} \equiv \sum_{i,j} v_{SC}(ij) w_{ij}^v \tag{6}$$

where $v_{SC}(ij) = \mathbf{u}_{ij}^{LoS} \cdot \mathbf{v}_{SC}$ is the projection of the satellite velocity along the line-of-sight axis.

### 2.1.1  Surface DEM and visibility algorithm

The Advanced Spaceborne Thermal Emission and Reflection Radiometer (ASTER) Global Digital Elevation Model (GDEM) (https://asterweb.jpl.nasa.gov/GDEM.asp, Abrams et al. (2010)) provides finely resolved ($1'' \times 1''$, i.e. 30.9 m $\times$ 30.9 m at the equator) global topography maps. In this work, we focused on the Piedmont region, which is located in the NW of Italy and is of particular interest for the orography associated with the Western Alps on the border with France. From the geolocated elevation data it is possible to derive useful quantities such as the distance between the satellite and the different elementary surfaces $dS_{ij}$ and the corresponding unit vector $\mathbf{u}_{ij}^{LoS}$ but also the two other unit vectors $\mathbf{u}_{ij}^n$ and $\mathbf{u}_{ij}^z$ previously defined (see inset in Fig. 2). Each pixel position, defined by the north-west vertex, is identified with latitude, longitude and altitude (LLA) coordinates in the WGS84 reference frame, and can be transformed from LLA to cartesian ECI coordinates $r_{i,j}^{ECI}$; for this step, assumption of a spherical Earth is used. The normal to each pixel, in general pointing outwards from the Earth surface, is found with the following relation:

$$\mathbf{u}_{i,j}^n = \left( \frac{\mathbf{r}_{i+1,j}^{ECI} - \mathbf{r}_{i,j}^{ECI}}{\|\mathbf{r}_{i+1,j}^{ECI} - \mathbf{r}_{i,j}^{ECI}\|} \right) \times \left( \frac{\mathbf{r}_{i,j+1}^{ECI} - \mathbf{r}_{i,j}^{ECI}}{\|\mathbf{r}_{i,j+1}^{ECI} - \mathbf{r}_{i,j}^{ECI}\|} \right)$$

with the indices $i,j$ ordered, respectively, from north to south and from west to east.

In the integrals of Eqs. (4-6) pixels that are visible must be identified. Paths of the electromagnetic radiation propagating from the radar in all different directions within the antenna pattern are indicated as rays, and are assumed to be straight lines since bending is negligible at these viewing angles (Fabry, 2015). Visibility is checked iteratively for each ray connecting

the spacecraft to the pixels of the considered DEM portion, up to a maximum altitude (which for this study was set to the maximum DEM regional value, 4564 m). Starting from each pixel and following such rays, range is decreased in small steps; then, altitude at the considered point is compared to the value obtained from interpolation of the DEM at the same horizontal coordinates. If the former is larger than the latter then the next iteration is performed; otherwise the visibility status is set to false and the iteration is aborted (see red ray in Fig. 3). If the maximum altitude is reached the visibility status is set to true and the ray tracing is terminated (see green ray in Fig. 3).

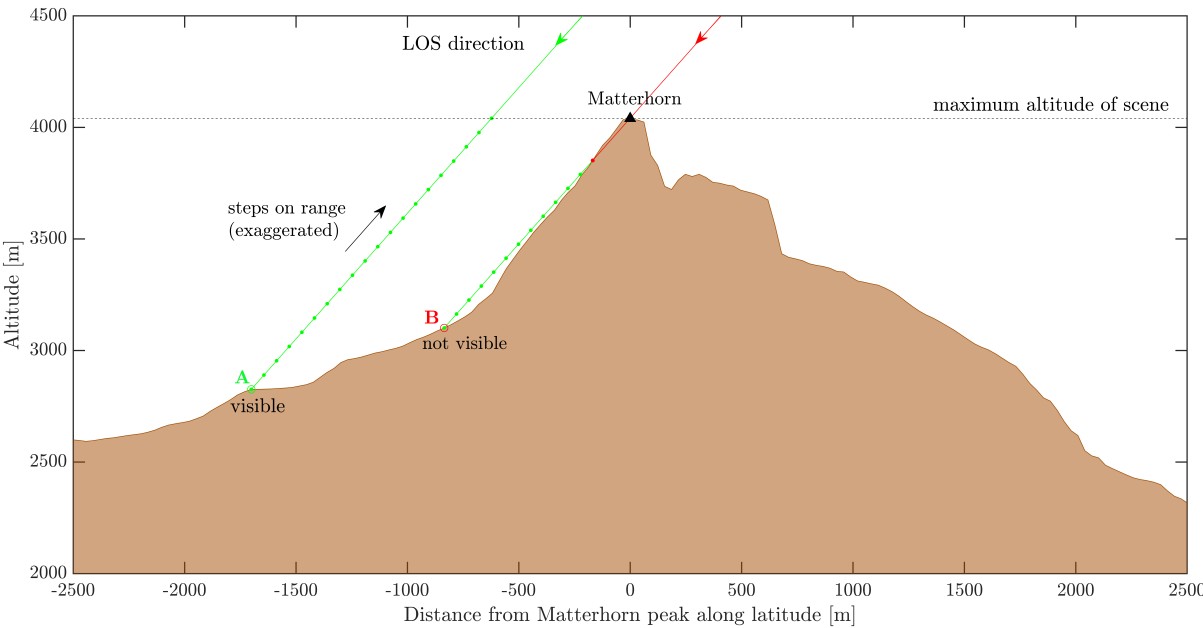

**Figure 3.** Schematic describing the idea underpinning the visibility algorithm with a longitudinal slice of a scene around the Matterhorn at a constant latitude of 44.667° E. For illustration purposes a red and a green ray are traced in proximity to the Matterhorn. Correspondingly point A (B) is (is not) visible. The dots correspond to the iterations done to check the visibility.

## 2.2 Terrain classification and NRCS

The other key element in the integrals of Eqs. (4-6) are the NRCSs. Ground-based field campaign measurements in the 1980s for different land surfaces (Ulaby and Dodson, 1991) and more recent airborne measurements over water bodies (Battaglia et al., 2017; Wolde et al., 2019) have been used to create look-up-tables (LUTs). Seven surface types have been selected as representative of different NRCS behaviour (see list in Tab. 2) according to the available LUTs.

**Table 2.** List of surface types with an available NRCS model. Water is added to the six land surface categories present in the NRCS database as parameterised in Ulaby and Dodson (1991).

| Number | Name |
|--------|------|
| 1 | urban |
| 2 | grass/short vegetation |
| 3 | water bodies |
| 4 | trees |
| 5 | soil, rocks |
| 6 | snow/ice |
| 7 | shrubs |

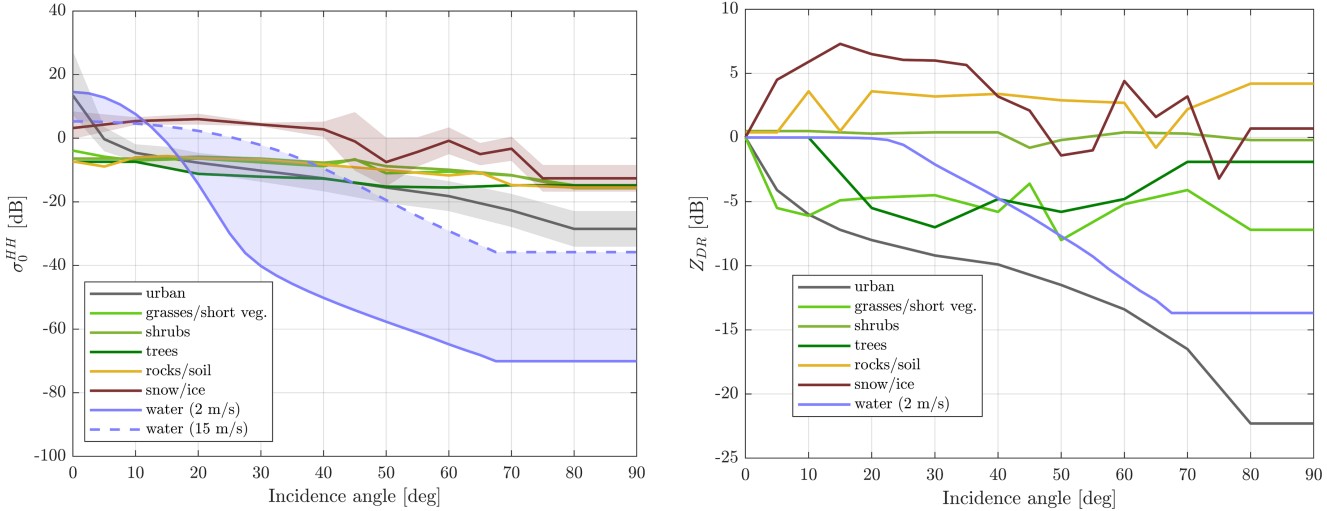

**Figure 4.** NRCS for H-polarised radiation, $\sigma_0^{HH}$ (left) and linear depolarisation ratio $Z_{DR} \equiv \sigma_0^{HH} - \sigma_0^{VV}$ as a function of the incidence angle for the different surface types as used in this study (see Tab. 2). For two of the surface types the shaded area indicates the observed standard deviation.

The dependence of the NRCS for H-polarised radiation, $\sigma_0^{HH}$ (left) and of linear depolarisation ratio $Z_{DR} \equiv \sigma_0^{HH} - \sigma_0^{VV}$ as a function of the incidence angle are shown in Fig. 4. Few remarks:

1. water surfaces have a strong dependence on the incidence angle with very strong surface dimming when moving towards high incidence angles;

2. land surfaces (with the exception of urban surfaces) show a much flatter response of the NRCS with the incidence angle
with slight decreases with increasing incidence angles;

3. at about the WIVERN incidence angle land and ocean NRCSs vary in the range between 5 and -25 dB and between -15 and -50 dB, respectively;

4. close to nadir NRCSs vary broadly in the range between -10 and 20 dB in rough agreement with CloudSat measurements (Durden et al., 2011).

 5. $Z_{DR}$ are usually negative with few positive values in correspondence to rocks/soils and snow/ice.

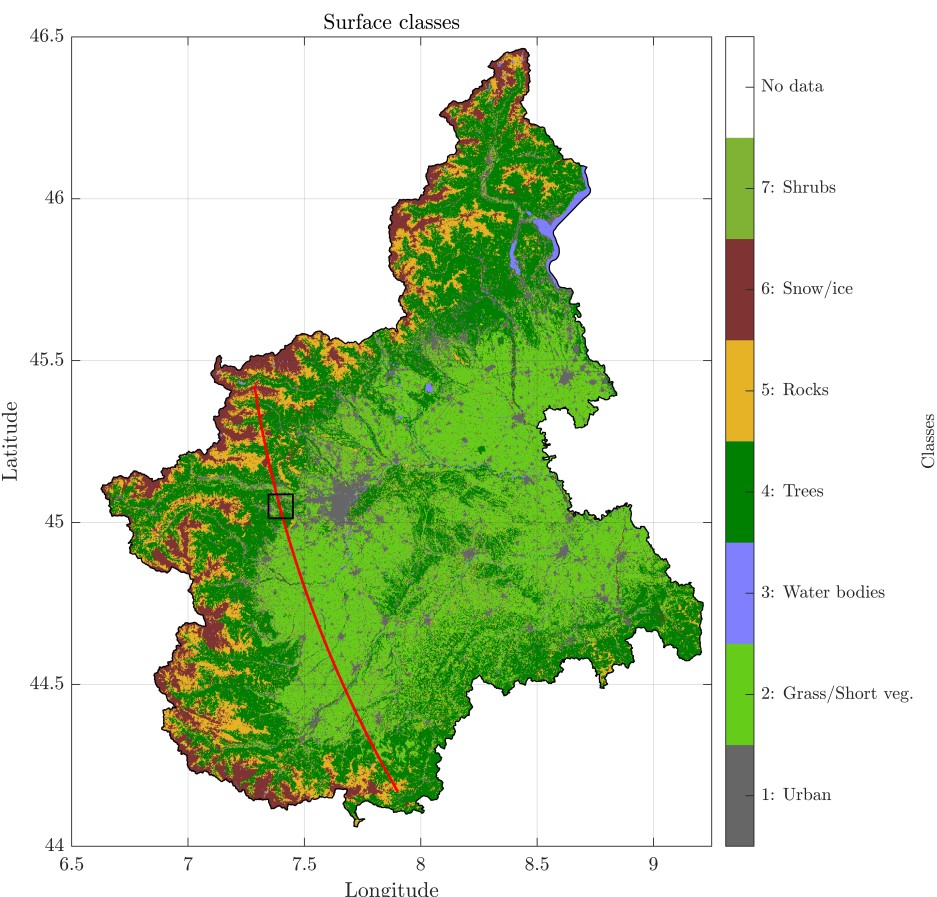

**Figure 5.** Terrain characterization of the Piedmont area (courtesy of Stefano Campus, GEOPIEMONTE). The red line represents the ground track of the antenna boresight for a case study scan (see Sect. 3.2). The black square corresponds to the region used in the single footprint case study in Sect. 3.1.

A detailed surface classification map of the Piedmont area at 20 m resolution with dozens of classes has been provided by GEOPIEMONTE (https://geoportale.igr.piemonte.it/cms/). These terrain categories have been mapped into the seven classes listed in Tab. 2 for which a NRCS model was available and interpolated in the same grid as the DEM. The results for the Piedmont region is depicted in Fig. 5. Note the mountainous regions in the Western part at the border with France with mainly

rocks, trees and snow/ice. This classification linked with the LUTs described in Sect. 2.2 allows computing the co-polar NRCSs at any given angle for H and V-polarised radiation.

## 2.3    Inclusion of noise and receiver response function

Once the ideal reflectivity, Doppler velocity and Doppler width profiles are computed according to Eqs. 4-6 at a range resolution of 50 m, real Doppler and reflectivity signals are generated according to the method proposed by Battaglia et al. (2022b) and

further improved in Battaglia et al. (2024). This takes into account the polarisation diversity (PD) (Battaglia et al., 2013) pulse sequence envisaged for WIVERN (Illingworth et al., 2018) with H and V pairs closely transmitted (with a separation of $20\,\mu s$) and with PD pairs transmitted every $250\,\mu s$ and the assumption that the pair repetition time is larger than the decorrelation time so that only pulses within the same polarisation diversity pair are correlated. The H and V pulses in each pairs have correlations computed in the approximation that spectra are Gaussian with the given mean Doppler velocity and spectral width

(Pazmany et al., 1999). Noise corresponding to a single pulse -18 dBZ equivalent reflectivity is added to the signal. The I and Q are sampled every 50 m in range. Then they are convolved with a Hamming window to simulate the receiver response (Schutgens, 2008). Finally polarisation diversity pulse pair (PDPP) estimators (Battaglia et al., 2013, 2024) are used to compute the reflectivity and the mean Doppler velocity profiles.

    An important consideration. Land surfaces are generally characterised by large values of linear depolarisation (-10 to -3 dB)

and low values of $\rho_{HV}$, the correlation between H and V polarized signals (0.4 to 0.8). While there is not much correlation for the co-polar surface signals there is an excellent correlation between the cross-polar signals generated by the surface (the so-called "surface ghosts" as discussed in Illingworth et al. (2018); Rizik et al. (2023) which appear above and below the surface and are separated in range by $2\Delta r_{T_{HV}} = 4cT_{HV}$ (Battaglia et al., 2024). These signals can then be used to extract the Doppler signal by performing a dedicated pulse pair processing that correlates the H and V profiles shifted by $2\Delta r_{T_{HV}}$. For

such Doppler estimate, the reduction in signal-to-noise ratio ($SNR$) associated with the surface linear depolarisation ratio is well compensated by the improvement in the Doppler estimators associated with the substantial increase in correlation. In the following two cases are considered:

1. a low correlation case ($\rho_{HV} = 0.5$) but with the SNR expected from the $\sigma_0$ of the surfaces, representative of the standard PDPP processing;

2. a high correlation case ($\rho_{HV} = 0.98$) but with the SNR reduced by 5 dB compared to the $\sigma_0$ of the surface in order to account for the cross-talk. This case is representative of the Doppler estimates obtained by correlating the ghost signals.

These two cases are better illustrated in Fig. 6. The two co-polar signals at the surface range correspond to the H (first) and V (second) separate pulses sent $T_{HV}$ apart, reflecting back from the surface without changing polarization. The two cross-talk signals originate from the same H or V pulses which are backscattered in the cross polarization and therefore appearing at

different ranges (higher above or below the surface). For these signals the return power is lower but the correlation is much higher because of electromagnetic reciprocity.

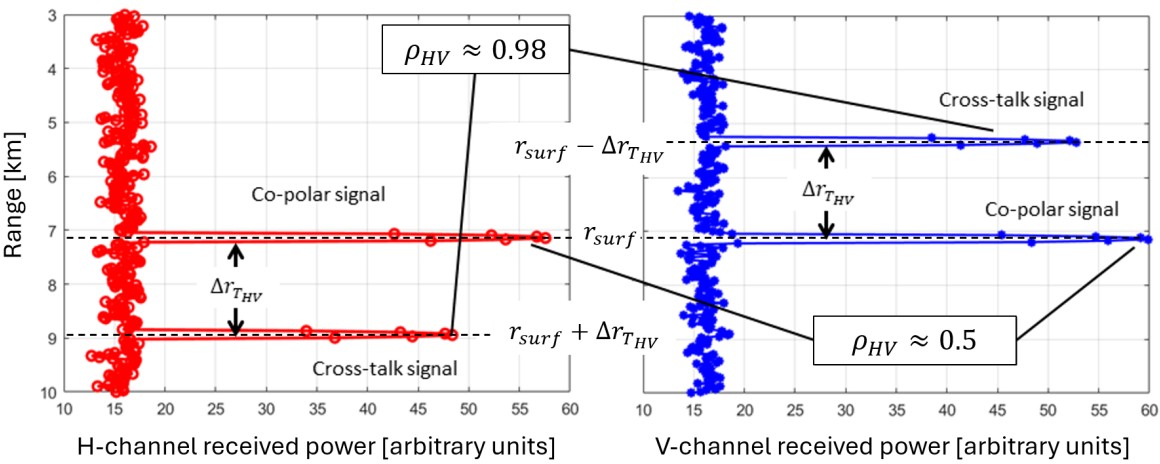

**Figure 6.** Schematic describing the two possible methods for deriving surface Doppler based on pulse pair estimates (the black lines indicate the signals that are correlated).

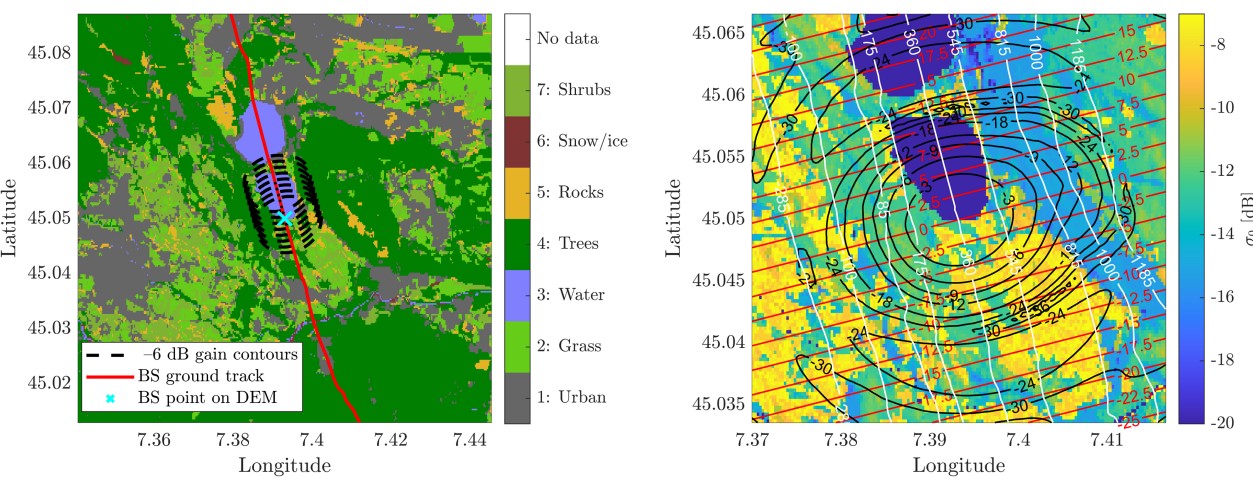

**Figure 7.** Case study with the WIVERN scanning ground track (red line) passing through the Western part of Piedmont region (see Fig. 5) with details of the terrain classes (left panel) and NRCS values (right panel). The profile with radar antenna boresight intercepting the DEM surface at the cyan cross near the Avigliana lakes (blue patches) is discussed in detail in the text. The right panel depicts all the relevant quantities that enter Eqs. (4-6) with the iso-range contours in white (converted to heights above the geoid), the iso-Doppler contours in red and the antenna iso-gain contours in black and the NRCS (colour-coded); the colorbar has been clipped to a minimum value of -20 dB, but values over the two lakes are very low (around -50 dB). Contours where the antenna gain is 6 dB lower than the maximum gain corresponding to the 8 footprints used to compute the 1 km averaging around the cyan cross are also shown in the left panel.

# 3 Case studies

## 3.1 Single footprint

A single footprint scene has been chosen as case study to illustrate the effect of NUBF (Figs. 7, 8, 9). The scene, with the radar
in left side-looking configuration (antenna rotation angle $\phi_A$, or azimuth, of about $90°$, measured counter-clockwise from the satellite track direction), is centred over the Avigliana lakes, with the antenna boresight (bs) hitting the southern shore of the southernmost lake (cyan cross in the left panel of Fig. 7). The ideal reflectivity and Doppler profiles (no noise, no receiver added) are shown in Fig. 8. Sidelobes contributions are included up to -30 dB (see Fig. 7 and 9).

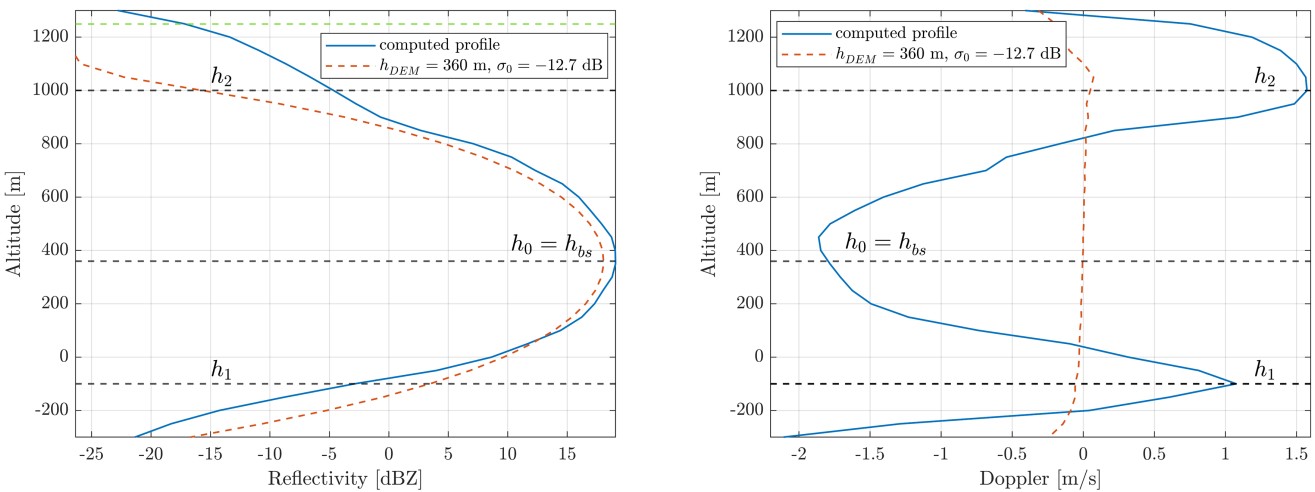

**Figure 8.** Case study for NUBF: reflectivity and Doppler profiles. Three heights have been selected, corresponding to peaks in the Doppler velocity profile (dashed horizontal black lines). The $h_0 = h_{bs}$ corresponds to the height of the point hit on the DEM by boresight axis (360 m). Ideal profiles in case of constant DEM height and constant NRCS equal to the local means are included. The green dotted line in the top of the left panel represents the height at which the profile drops below the single-pulse minimum detectable reflectivity of -18 dBZ, which corresponds to 1249 m in this case. Subtracting the boresight height (360 m), a -18 dBZ clutter depth of 889 m is obtained for this scene. This quantity will be explained in Sect. 4.1.

The reflectivity peaks at about 19 dBZ at an height of 360 m, where the boresight hits the ground, and then decreases below
the -18 dBZ noise level at a height of about 1260 m and -250 m. Due to the presence of higher $\sigma_0$ values in correspondence to iso-ranges with $h > 360$ m, the clutter is more pronounced at ranges smaller than the boresight range (see right panel in Fig. 7). The Doppler profile on the other hand presents a very anomalous behaviour compared to the flat homogeneous terrain reference case (red dashed line). In correspondence to the boresight height $h_0$, a negative Doppler velocity of -1.85 m/s is simulated. Near the centre of the beam, where the maximum antenna gain is achieved, pixels with negative Doppler velocities on the
bottom half of the scene (see red contour lines) present higher NRCS than the ones in the top half (see Fig. 7). The calm water body in this part of the scene presents $\sigma_0$ values around -50 dB at about $41.6°$ incidence (see Fig. 4), resulting in a negative

Doppler bias around $h_0$. The two positive peaks observed at different altitudes ($h_1$ and $h_2$ in the right panel of Fig. 8) can be explained considering, on top of the $\sigma_0$ variability, the orography, which distorts the iso-range lines and the gain pattern on the ground. The principle is better outlined considering Eq. (6), which highlights that the Doppler velocity at a given range is the result of the weighted average, for all surface domains corresponding to that given range, of the satellite velocity projection along the line-of-sight (value in the red iso-lines in Fig. 9) with the weights $w_{ij}^v$. The weights correspond to the return power reflected to the radar by a given surface pixel, which results from a combination of NRCS and the square of the antenna gain. In Fig. 9 these weights are depicted for the three annuli that constitute the area of integration for the three chosen heights. In correspondence of the two Doppler profile maxima ($h_1$ and $h_2$ in Fig. 8), pixels with positive velocities are characterised by higher antenna gain values than pixels with negative velocities; this converts in larger weights $w_{ij}^v$ for $v_{SC}(ij) > 0$ and thus positive velocities overall.

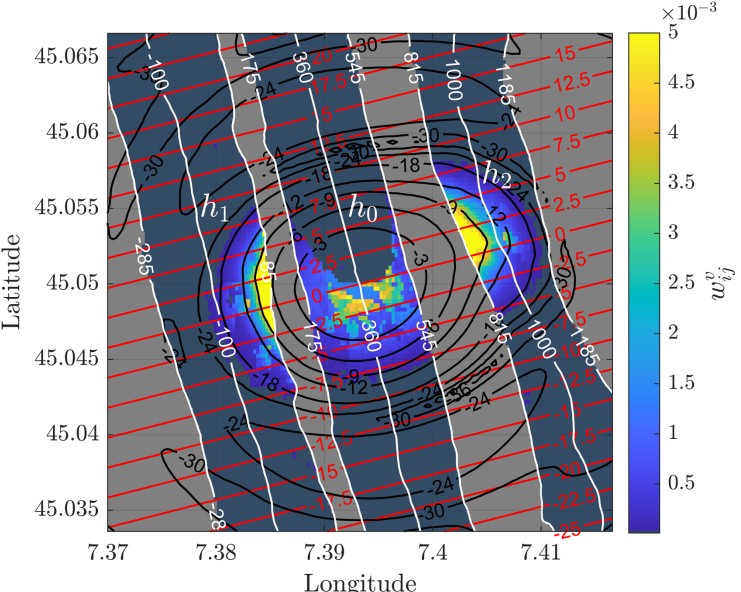

**Figure 9.** Case study for NUBF: map of the weights given to each pixel for the computation of the Doppler velocity at a given height (see Eq. 6). The weights are defined as $w_{ij}^v = C_S P_r^{-1}(r_{ij}) \ \sigma_0(\psi_{ij}) \ G_n^2\left(u_{ij}^{LoS}\right) \ r_{ij}^{-4} \ dS_{ij}$. Three heights corresponding to the peaks in the Doppler velocity profile have been selected; the three highlighted regions correspond to the surface domain that contributes to the integral in Eq. (6) for each of the the chosen heights (i.e. corresponds to ranges equal to the three given ranges $\pm 250\,m$). Values fading to grey inside the annulus have magnitude lower than $10^{-5}$ and therefore can be neglected.

In summary, two different effects can be appreciated due to orography and $\sigma_0$ variability. Along the central annulus, the water body has very low NRCS in the regions with positive Doppler values, resulting in a negative velocity bias. Inside the two lateral annuli of Fig. 9, the surface orography distorts the iso-range lines resulting in higher gain values for pixels with positive velocity, with a positive bias at the corresponding heights.

An additional observation can be made: in Fig. 8 the real reflectivity profile in blue does not show much variation with respect to the red dashed line (although some change in the power distribution can nevertheless be seen). This can be explained by looking at Fig. 9 and the right panel of Fig. 7: the $\sigma_0$ gradient is roughly parallel to the iso-range lines, so NUBF does not affect the reflectivity integration inside each annulus. More variability would have been obtained if the same scene were looked at in a forward or backward view at an azimuth of $0°$ or $180°$.

## 3.2  Scan

WIVERN (see Specs. in Tab. 1) will perform conical scans, moving the footprint at approximately 500 km/s, and transmitting 8 pairs of H and V pulses every km. Consequently, each footprint will be spaced by roughly 125 m along the boresight scanning track (e.g., see the black lines in the left panel of Fig. 7). The methodology described in Sect. 3.1 can be repeated, incorporating noise as outlined in Sect. 2.3, and the results can then be averaged over an arbitrary distance. Figure 10 shows the results of a single scan in side configuration across the western part of the Piedmont region, covering a total length of approximately 150 km (left panel). The reflectivity and Doppler velocity profiles (top two right panels) are averaged over every 8 pulses, corresponding to an integration length of roughly 1 km. Here, the high correlation estimator is selected.

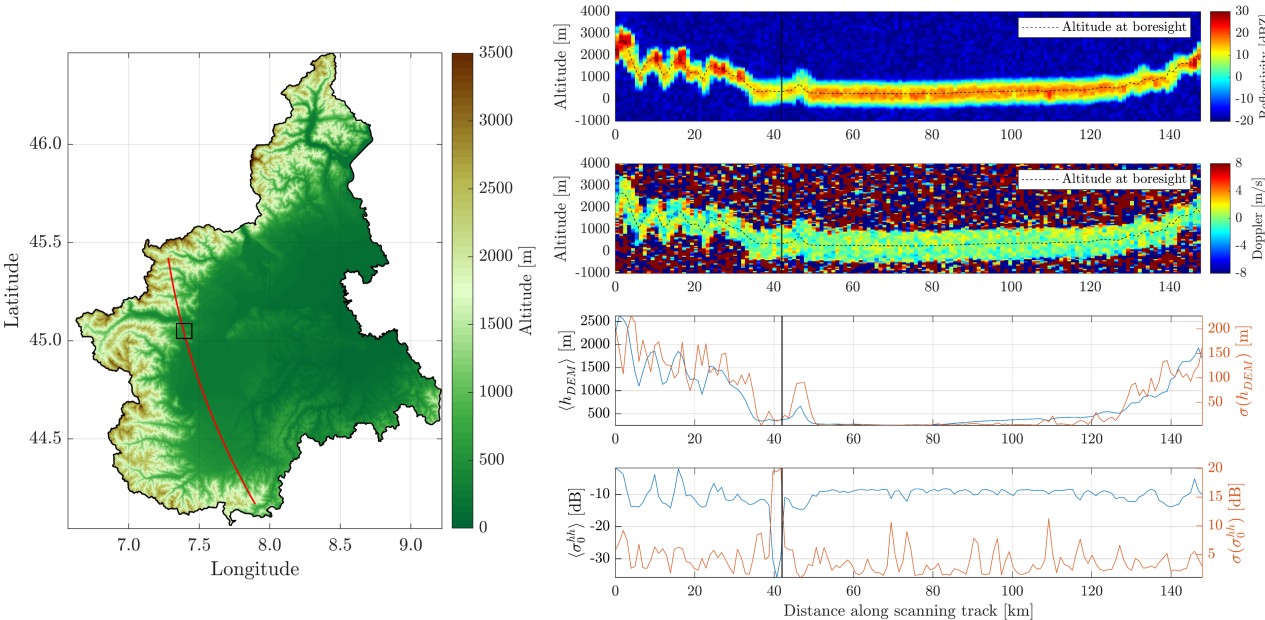

**Figure 10.** Case study of a continuous scan in side configuration, with an azimuth of around $90°$. Left panel: the ground track of the antenna boresight (red line) with a counter-clockwise scanning across the Western Piedmont region with colours modulated by the orography. The black square corresponds to the region used in the single footprint case study (Sect. 3.1). Right panel: reflectivity (first row), Doppler velocity (second row), mean and standard deviation (respectively, blue and red lines), across the 1 km averaging region, for elevation (third row) and NRCS (fourth row). The black vertical continuous lines represent the position of the single footprint case study seen in Sect. 3.1.

The black dashed line in the two top right panels represents the height of the point where the boresight intersects the DEM surface, averaged over 8 samples. The Doppler profiles have been clipped to $\pm 8$ m/s to highlight the presence of noise and deviations in the otherwise nearly flat profiles. The mean and standard deviation values of DEM elevation and NRCS are also provided (two bottom-right panels). These values are calculated by considering a $1 \times 1$ km square subgrid, centred on the boresight axis and averaged every 8 samples along the ground track. These values are later used in the statistical analysis in

Sect. 4, which considers a large number of scans similar to the one presented here.

In general, it can be observed that the reflectivity peaks closely follow the $h_{bs}$ values (black dashed line in the top-right panels), which are also close to the DEM elevation mean values $\langle h_{DEM} \rangle$ (blue line in the third-right panel), except in areas with pronounced orography (e.g., the first 30 km of the scan). The clutter return generally falls below the single-pulse minimum detectable reflectivity of -18 dBZ at approximately $\pm 900$ m from $h_{bs}$.

Outside these heights, the Doppler signal becomes increasingly noisy, practically reducing to a random number inside $\pm$ the Nyquist velocity of about 40 m/s at low SNR. Regions with significant deviations from the ideal flat reflectivity and Doppler profile correspond well to regions with higher standard deviation in DEM height and NRCS. In orographic regions at the beginning and end of the scan, the reflectivity profiles deviate notably from the flat homogeneous shape, unlike the profiles in the middle segment, originating from a flatter portion of terrain.

## 260    4    Statistical results

In order to capture a wide range of samples with different orography and inhomogeneity conditions, a large number of scans have been performed over the Piedmont region; along-track averages over 1 or 5 km (respectively 8 or 40 samples) have been performed. Each average produces a reflectivity and Doppler velocity profile in correspondence with a DEM elevation and NRCS mean and standard deviation value and an azimuth scanning angle.

### 265    4.1    Doppler velocity at boresight altitude: departures from 0 m/s

The boresight Doppler velocity value $v_{D,bs}$ is found as the value nearest to the height of the point hit by the antenna boresight. In presence of flat terrain and homogeneous surfaces, this value is expected to be zero, but noise, orography, and NRCS variations across each footprint introduce departures. Four different cases have been studied, based on two different averaging distances (1 or 5 km) and two $\rho_{HV}$ values (0.98 or 0.5). The larger the variability of the DEM height and NRCS within the

averaging domain, the larger the departure from the 0 m/s reference. Therefore, by clustering profiles based on four standard deviations of the DEM height [1) $\sigma(h_{DEM}) < 10$ m; 2) $10$ m $\leq \sigma(h_{DEM}) < 50$ m; 3 ) $50$ m $\leq \sigma(h_{DEM}) < 150$ m; 4) $150$ m $\leq \sigma(h_{DEM}) < 623$] and four NRCS standard deviations [1) $\sigma(\sigma_0) < 3$ dB; 2) $3$ dB $\leq \sigma(\sigma_0) < 5$ dB; 3) $5$ dB $\leq \sigma(\sigma_0) < 7$ dB; 4) $7$ dB $\leq \sigma(\sigma_0) < 22$ dB], 16 different classes have been identified and histograms for $v_{D,bs}$ for each class have been built. Results are reported in Fig. 11 for the four cases. A few considerations can be drawn.

–    The classes have been chosen to include a significant number of occurrences ($N$ inside the boxes), but, as the terrain in the chosen region is relatively flat, in general classes with smaller standard deviation in elevation present more occurrences.

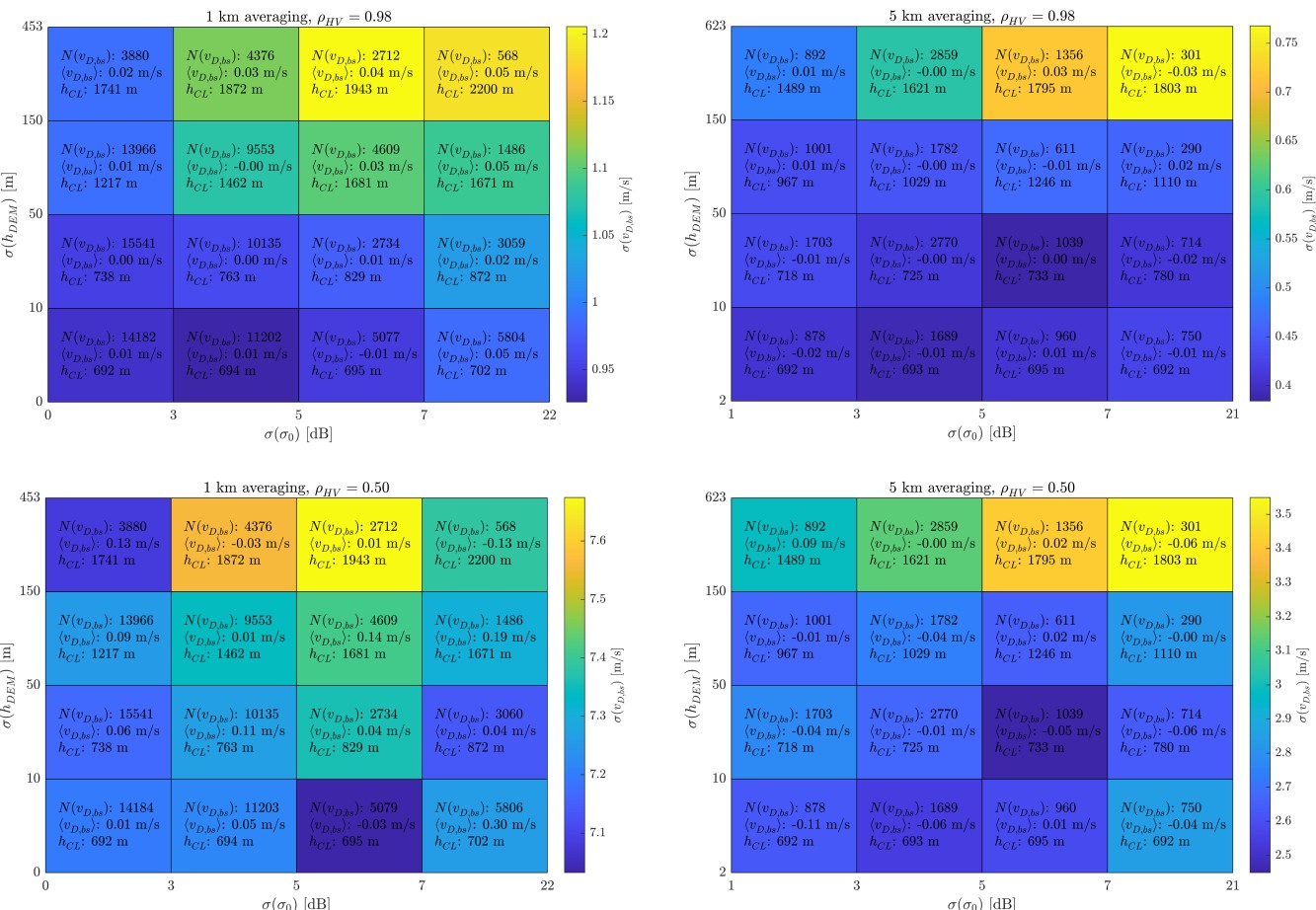

**Figure 11.** Boresight Doppler velocity statistical analysis for the four different cases: 1 km averaging, $\rho_{HV} = 0.98$ (top left); 5 km averaging, $\rho_{HV} = 0.98$ (top right); 1 km averaging, $\rho_{HV} = 0.5$ (bottom left); 5 km averaging, $\rho_{HV} = 0.5$ (bottom right). For each box corresponding to one of the 16 classes the colour indicates the $\sigma(v_{D,bs})$ value. Inside each box, the number of occurrences $N(v_{D,bs})$, the mean value $\langle v_{D,bs} \rangle$, and the 95th percentile value of the clutter depth height $h_{CL}$ are shown. The $h_{CL}$ for the -18 dBZ level values have been computed using the averages of the ideal profiles rather than the noisy ones.

– As expected, because of the different viewing geometry included in the database, the mean values of all the histograms, $\langle v_{D,bs} \rangle$, is close to 0 m/s for all classes. This confirms the fact that the surface can be used for calibrating the Doppler signal but in some cases only after substantial averaging.

– The standard deviation of the histograms, $\sigma(v_{D,bs})$, generally increases when moving towards higher DEM height and NRCS standard deviations.

– In presence of almost flat and homogeneous surfaces (bottom left pixels) $\sigma(v_{D,bs})$ is dominated by the noise. This baseline value heavily depends on the correlation $\rho_{HV}$ and the averaging distance. $\rho_{HV} = 0.5$ produces extremely noisy

Doppler velocities with a baseline exceeding 7 m/s. Only after 5 km averaging this can be brought down to 2.5 m/s. The high correlation value $\rho_{HV} = 0.98$ (which implies getting the surface Doppler velocity via the ghost processing) produces much better results and seems very promising.

- The effect of orography and NRCS inhomogeneity can be seen only when moving towards large values of higher standard deviations, while at lower standard deviations noise dominates and essentially defines the lower bound for $\sigma(v_{D,bs})$.

## 4.2 CFAD

Contour Frequency Altitude Display (CFAD) plots have been computed to show the variability of the reflectivity and Doppler velocity profiles when having a given variability of the DEM and NRCS across the integration zone (Fig. 12). For illustrative purposes, the case with $\rho_{HV} = 0.98$ and averaging over 1 km has been chosen for only two classes: 1) in the left column, almost perfectly flat regions with homogeneous NRCS (corresponding to the bottom left box of panels in Fig. 11); 2) in the right column, regions with very mountainous terrain and strong inhomogeneous NRCS (corresponding to the top right box of panels in Fig. 11). For Doppler velocities, a division based on $\phi_A$ was adopted, as the shape of the profiles change based on the antenna rotation angle. All CFADs have been rescaled to a renormalised height so that $h_{bs}$ is set to 0 m, by subtracting from the height of each profile the height at which the boresight axis intercepts the DEM surface.

In general, for almost flat and homogeneous surfaces (left panels), envelopes are more compact and they tend to behave similarly to the perfectly flat terrain (Scarsi et al., 2024) with the characteristic shape of the Doppler profiles for azimuth near to forward/backward (centre left panel) and side view (bottom left panel) whereas when increasing DEM elevation and NRCS variability (right panels), profiles present larger spread. For instance, in the first row, reflectivities drop below the single-pulse minimum detectable value of approximately -18 dBZ at about 700 m in the left panel rather than at much larger values in the right panel. For the different classes, $h_{CL}$ the height above surface at which 95% of the clutter profiles have reflectivity (without any noise) lower than -18 dBZ has been computed and is indicated as the third number in the boxes of Fig. 11. Results clearly show how the clutter region moves from 700 m in presence of flat terrain to more than 2 km in complex orography conditions.

Similarly, the inclination of the Doppler profiles near forward/backward looks (second row) becomes less pronounced but more scattered in presence of orography and NUBF.

## 5 Summary and conclusions

In this study, a novel simulator was developed to reproduce the clutter reflectivity and the Doppler velocity signal as expected for a spaceborne scanning Doppler radar instrument. The simulator is based on the ray-tracing approach with surface properties (slope, elevation, NRCS) derived from a high-resolution raster DEM and land classification map. A look-up table based on ground-based measurements is used to compute the normalised radar cross section (NRCS), $\sigma_0$. The clutter simulator has been applied to the WIVERN mission, one of the two remaining candidates within the ESA's Earth Explorer 11 programme, which

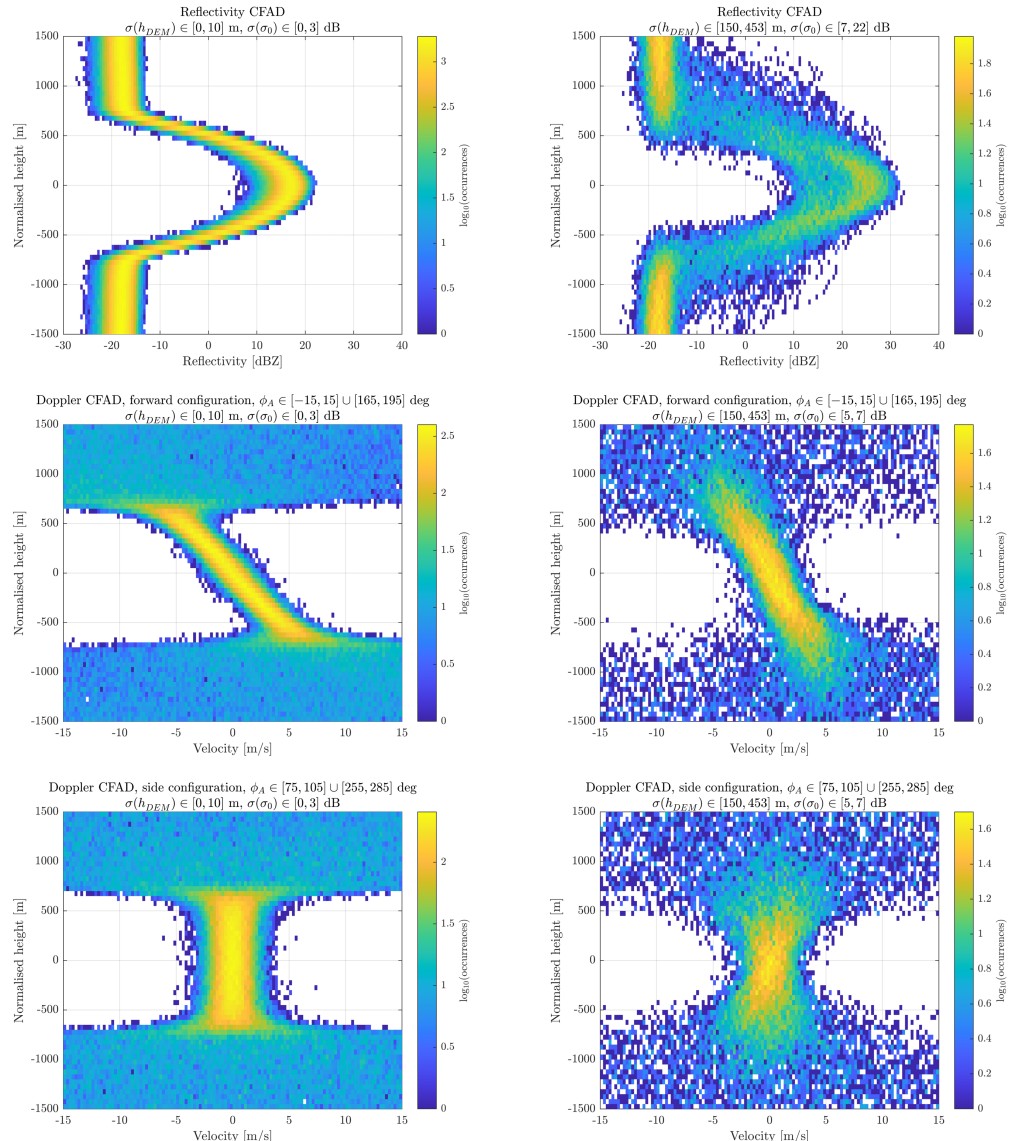

**Figure 12.** CFADs for reflectivity and Doppler, height is normalized by subtracting the boresight height $h_{bs}$, which is the height of the intersection between the boresight axis and the discretised surface defined by the DEM. On the left column we have the CFADs for the lowest standard deviation class, on the right for the highest. Top row: reflectivity CFADs. Middle row: Doppler velocity CFADs for forward configurations (profiles around $\phi_A = 180°$ are grouped together with those around $\phi_A = 0°$ changed in sign). Bottom row: Doppler velocity CFADs for side configuration.

proposes the use of a conically scanning W-band Doppler radar to study in-cloud winds and the micro- and macro-physical properties of clouds and precipitation. This works expands on the existing end-to-end simulator which simulates the radar

observations of atmospheric and surface target by using outputs from numerical weather prediction models, expanding on the currently simple implementation of the ground clutter signal.

The simulator allows the characterisation of the expected ground return over regions with known terrain characteristics. In this study, an example of application is shown over the Piedmont region of Italy, which offers a variety of different scenes due to the presence of the Alps to the north and west and the flat regions of the Po valley. The presence of surface orography and the inhomogeneity of the backscatter cross sections within the radar footprint cause significant deviations from the reference provided by a homogeneous and flat surface. These effects have been demonstrated by the choice of a case study over a lake shore with nearby orography, where the NUBF phenomenon could be discussed in detail.

Furthermore, the simulator has been used for statistical analysis to examine the effect of elevation and NRCS variability over a large number of scans. In particular, departures from the 0 m/s Doppler velocity at boresight have been discussed as a function of the integration length and the variability of $h_{DEM}$ and $\sigma_0$ within the integrating region. These results can be used to better assess over which regions and over which integration length the surface Doppler can be exploited for Doppler calibration purposes. Indeed, these aspects are of great importance for mispointing corrections of the Doppler signal. Data-driven calibrations using surface Doppler velocity measurements as an external calibration reference are the most effective, as demonstrated from ongoing work on the EarthCARE CPR. Our study has quantified the expected standard deviation of the surface $v_D$, for given NRCS and elevation variabilities and for a given integration length. Such values can be used to evaluate the use of such surfaces as reference points for Doppler calibration. This methodology has already proven very valuable for the calibration of the recently launched EarthCARE Doppler radar.

Also they demonstrate that, over relatively flat vegetated surfaces, the clutter reflectivity of the surface can remain below low reflectivities ($< -20$ dBZ) for all heights more than 1 km above the ground. The situation becomes much worse over mountainous ranges and in presence of rocks and bare soil.

Future work should address improvements to the $\sigma_0$ dataset as a function of incidence angle and land type; additional field campaign measurements with ground-based radars are strongly recommended. The NRCS dataset used to build the LUTs in Fig. 4 is based on experimental campaign carried out in the 1980s. Higher incidence angles are missing (the sampled incidence angles are also sparse) and for some class terrains the Ka or Ku band values had to be used, due to missing data in the W band. For systems adopting high incidence angles like WIVERN it will be critical to better establish the drop in NRCS when moving from nadir to very slant angles on surfaces covered with different types of snow, sea ice or different land biomes.

Further applications of this tool are possible also for missions with nadir-pointing radar instruments, as for the EarthCARE and CloudSat Cloud Profiling Radars (Tanelli et al., 2008; Illingworth et al., 2015; Kollias et al., 2023), or cross-track scanning, as for the Global Precipitation Measuring Dual Precipitation Radar (Skofronick-Jackson et al., 2016). For such systems, detailed simulations of the ground clutter and consequent refinement of clutter removal algorithms could pave the way to a better understanding of near-surface hydrometeor processes (e.g. orographic precipitation).

*Code and data availability.* Simulator code and all raw data are available upon request. The ASTER GDEM dataset is freely available subject to registration from NASA servers (https://asterweb.jpl.nasa.gov/GDEM.asp, Abrams et al. (2010)).

*Author contributions.* FM performed most of the simulations and the analyses. AB contributed to the analysis, the writing and has defined the project. PK contributed to the discussion and the review of the paper.

*Competing interests.* At least one of the (co-)authors is a member of the editorial board of *Atmospheric Measurement Techniques*.

*Acknowledgements.* This research has been supported by the European Space Agency under the activities "WInd VElocity Radar Nephoscope (WIVERN) Phase A Science and Requirements Consolidation Study" (ESA Contract Number RFP/3-18420/24/NL/IB/ab) and by the Italian Space Agency (ASI) project "Scientific studies for the Wind Velociy Radar Nephoscope (WIVERN) mission" (Project number: 2023-44-HH.0). This study was carried out within the Space It Up project funded by the Italian Space Agency, ASI, and the Ministry of University and Research, MUR, under contract n. 2024-5-E.0 - CUP n. I53D24000060005. This research used the Mafalda cluster at Politecnico di Torino.

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
