# Peer review of "Characterization of surface clutter signal in presence of orography for a spaceborne conically scanning W-band Doppler radar"

_EGUsphere, 2024_

## Referee Comment (RC2)

**Comments on 'Characterization of surface clutter ..' by Manconi et al., egusphere-2024-2779**

Simulators are useful for predicting the performance of instruments and helpful in understanding various error sources and in devising algorithms to extract the maximum amount of information. In this paper a simulator for the proposed WIVERN W-band radar is used to examine the behavior of the reflectivity and Doppler profiles over mountainous terrain.

Although I have several questions on the details, I found the paper informative. Since the WIVERN radar will be used primarily for cloud sensing, I expected to see some results on the atmospheric effects on surface cross section and Doppler but perhaps that will be dealt with in a separate paper. I recommend publication after the authors address the comments below.

Table 1: It's not clear to me whether the radar will transmit H and receive H and V (and transmit V and receive H and V) or whether it will transmit H and receive H only and transmit V and receive V only.

More generally, I'm not clear about the meaning of the 'ghost pulse' and how this affects the Doppler processing. This issue comes up later in the paper where the rho(HV) parameter, which implies that the cross-pol will be measured, is used to categorize the results. A few more sentences would be helpful to explain how this parameter is related to the scattering properties of the surface.

Fig. 4. The DPR (dual-frequency precipitation radar on the GPM satellite) surface cross-section data (Ku/Ka-band) over land shows a sharper decrease with angle in moving off nadir than the results shown here. The DPR data covers the angle range from nadir to 18 deg so airborne data are needed to fill in at the higher incidence angles.

I couldn't remember how sharp the drop-off with angle was but found the figure below. The land data are not categorized by surface type – all surface types are included.

The radar frequency is not mentioned in Fig. 4 and while the DPR database shows the Ku & Ka-band are similar in their angle dependencies over both land and ocean, the W-band data might depart significantly from the lower frequency data.

Unfortunately, there doesn't seem to be much off-nadir sigma-zero data at W-band, at least that I'm aware of. One advantage of the simulator is that the surface scattering model can be updated as new information becomes available.

[Figure]

Mean values of DPR sigma-zero from 1-month of data are shown on the left for ocean (top) and land (bottom). These data were taken from $35^0$ S to $35^0$ N to match the TRMM coverage. The Ka-band cross section over land decreases by about 12 dB going from nadir to $9^0$. (Since the data used here were measured early in the mission, the Ka-band data extended only to $9^0$.)

Lines 148-155. I had trouble following this discussion. First, it would be clearer to say something like: 'Land surfaces are generally categorized by large values of depolarization (-10 to -3 dB) and low values of rho(HV) (0.4 to 0.8).'

My confusion comes in the next sentence. 'While there is not much correlation for the co-polar surface signals ..' *(does this mean between rho(HH) and rho(VV))* '...there is an excellent correlation between the cross-polar signals ...'. – italics mine. But the previous sentence stated that these values are low so I must be missing something.

Is 'rho' in lines 157 and 159 the same as 'rho(HV)' in the previous paragraph? If so, the same notation should be used.

Fig. 6 caption: 'wo' → 'two'.

Line 166: I see a blue 'X' but not a black cross.

I'm a bit confused by Fig. 7 and the associated discussion. It is assumed that the antenna is pointed in a direction orthogonal to the satellite velocity vector so if the sigma-zero were uniform over the footprint, then the Doppler would be zero - as shown by the red line. The variation in range

in the Doppler is presumably caused by NUBF so a positive Doppler (assuming positive is toward the radar) would be caused by the return power from the forward portion of the beam being larger than that from the backward portion of the beam.  Is this correct?

I would have expected the reflectivity profile to be much more variable in range than the blue line shown in Fig. 7 left panel.  How typical is this; how much does it change when a field of view over the mountains is taken?

Not sure if side lobes are included in the antenna pattern but these would add to the Z variability, especially in the mountains.

Fig 9 caption on explanation of the bottom two panels, right-hand side.  Presumably, means are given by the blue lines and std dev's are given by the red lines.  This should be mentioned.

Does '1 km averaging region of elevation' mean that for calculation of the surface cross section the radar return power is used over a 1 km range window to compute mean and std dev?  For example, if the mean & std dev at a particular point are (15, 5) dB, does this mean that about 66% of the data falls within 10 to 20 dB?

Does the phrase 'with the antenna scanning at the side of the satellite ground track' mean that the data are taken at an azimuthal angle at 270 deg?

Line 209:  correspondence

line 213:  '.. this value is expected to be zero (*for fields of view orthogonal to the direction of spacecraft motion*)..' .   - italics mine.  Although this was noted earlier, I think it's important to emphasize that the direction along the incidence angle is perpendicular to the spacecraft motion.

Line 218:  should the fourth category be:

7 dB < std(s0) < 25 dB ?

Use of 'dB' here and use of 'meter' for the std dev of height might make this more readable and remind the reader of the units.

Line 220:  'A few ' rather than 'Few '.  'Few' implies 'Only a few'.

Lines 221-222: point 1 is difficult to understand and should be rewritten.

One possibility:  The classes have been defined to include a significant number of cases in each. Those classes where the standard deviation in elevation is small have a high-count number because much of the terrain in the segment chosen is relatively flat.

(The unwritten assumptions are that low std dev in elevation implies relatively flat terrain which implies a small standard deviation in sigma-zero.  But I think these assumptions are OK.)

Fig. 11:  I'm having trouble understanding the behavior of the Doppler in the middle figures.  From the title of the left middle figure, it seems that phi(A) is being varied from -15 to 15 deg but wouldn't the Doppler be the same regardless of a change of sign in phi(A)?  What parameter is being changed to produce the positive and negative Doppler.

For the forward-looking case, the large Doppler shift induced by the satellite motion has been subtracted off, correct?

Line 242: renormalised

---

## Author Response (AR1)

*We thank the anonymous referee for his review and his detailed comments.*

*Below, the referee's comments can be found in bold, followed by the authors' replies.*

**This manuscript describes development of a clutter simulator that was applied to the WIVERN mission to generate simulated reflectivity and Doppler velocity profiles over a mountainous region in the northwest of Italy. The manuscript presents detailed analysis of a particular profile that demonstrates non-uniform beam filling (NUBF) and also statistical analysis of a collection of scans over the region. The statistical results begin to quantify the intuitive idea that regions with greater variability in terrain elevation and NRCS will generate clutter profiles with greater variability in reflectivity and Doppler.**

**The authors state the aim of the work is to "extend the simulations of the clutter signal to non-planer surfaces, ... including a realistic variability of the surface backscatter" (lines 48-49). Their claimed novelty is "the application to a space-based configuration, the extension to the Doppler signal, and the inclusion of NUBF effects" (lines 51-54). The simulated data presented and analyzed appears to be high quality and analysis clearly illustrates NUBF and increased variability in reflectivity and Doppler profiles. The simulator considers only clutter (no atmosphere/hydrometeors simulated) and assumes no attenuation.**

**The work described in this manuscript is an incremental evolution of existing simulation techniques and the novelty is in the combination of high-resolution DEM with the WIVERN (W-band) mission. Other simulations with high-resolution DEMs have been previously described in the literature. The authors make no attempt to explain how the simulations would be useful for the WIVERN mission other than vague statements about how it may be difficult to use Doppler velocity as an antenna characterization technique over rough terrain (lines 10-11; 35-46; 223-225).**

Better understanding the shape of the clutter reflectivity and of the clutter mean Doppler velocity profiles is important for two reasons: 1) the reflectivity profile can be used for geolocation purposes and its shape is relevant for assessing the blind zone of a radar system (i.e. the region where the radar signal will not provide any useful information for the hydrometeors); 2) the surface Doppler can be used in a data-driven approach to mitigate mispointing errors associated to thermoelastic antenna distortions. This last aspect is being demonstrated by the EarthCARE CPR, currently in commissioning phase, which is using the ocean surface return as a reference for a data-driven mispointing calibration for distortions that vary along the orbit because of the changing sun illumination. The calibration of distortions that occur on shorter time scales needs more frequent calibration points. Therefore, it must be assessed what surfaces can be useful for this kind of purpose. The limits of acceptable variability in terms of sigma-zero and orography have to be addressed.

**There is no attempt to quantitatively link the simulated clutter Doppler profiles to potential errors in mispointing corrections or any other aspects of the mission. For these reasons it is difficult to see how the manuscript meets the criteria of scientific significance required by this journal.**

The errors in Doppler velocity measured at the surface at boresight can be converted to mispointing angle error (Scarsi et al., 2024). The mission requirements for the horizontal component of the line-of-sight wind measurements is in the order of 2.5 m/s: to achieve this goal, the contribution on mispointing errors must be lower than 0.4 m/s after all possible calibration methods (ESA-WIVERN-Team, 2023). The on-board attitude determination and control system can provide pointing of the antenna with an

uncertainty of the order of 1 m/s, thus not sufficient to meet the science requirements of the mission. In addition, thermoelastic deformation of the antenna is another large contribution to the pointing error, as confirmed during the commissioning phase of EarthCARE. This effect is cyclical with the orbital period and it's very hard to model and predict numerically, even using temperature sensors attached to the antenna. However, it produces a large effect on the mispointing of the boresight, even in the order of 1 m/s. For the WIVERN antenna thermoelastic deformations are expected to have the same magnitude (ESA, industrial studies). This effect must be corrected with additional methods, e.g. by looking into the surface Doppler velocity: calibration methods need natural targets, and the surface is the simplest one available.

**I suggest the authors make major revisions to this manuscript to address the issue of scientific significance before the manuscript is accepted for publication. In particular, the authors should make clear why it is a "substantial contribution" beyond existing simulation methods and make clear what benefits it will contribute to the WIVERN mission.**

To our knowledge, there is no clutter simulator of reflectivity and Doppler signal for spaceborne radars taking into account NUBF for ground return and orography that has been developed for the GPM, CloudSat, EarthCARE, or INCUS missions. So, we think this is a first. Of course it is more relevant for the WIVERN mission which has Doppler capabilities and is conically scanning.

This clutter simulator is a module of a larger end-to-end simulator endeavour being developed as part of the phase A activity funded by ESA, which simulates the full return from both atmospheric and surface targets, based on the simulator already developed at Politecnico di Torino (*Battaglia, A., Martire, P., Caubet, E., Phalippou, L., Stesina, F., Kollias, P., and Illingworth, A.: Observation error analysis for the WInd VElocity Radar Nephoscope W-band Doppler conically scanning spaceborne radar via end-to-end simulations, Atmos. Meas. Tech, 15, 3011–3030, 2022*; Rizik et al., 2023; Battaglia et al., 2024). This work completes the simulator adding a thorough treatment of the surface accounting for variability of sigma-zero at fine scales and orographic effects. In the previous simulator, the surface was treated in a simplistic way (flat and homogeneous) which is sufficient for oceanic surfaces. It is interesting, however, to understand land surface return to assess the surface Doppler for calibration and the relative strength of the signal from the clutter and from the atmosphere. Ongoing studies and private communications with the EarthCARE team (Pavlos Kollias and Bernat Puigdomènech Treserras) have highlighted the importance of such aspects.

Following these comments, revisions to the introduction and conclusion sections have been made to address the points made in the previous responses and to better illustrate the scope of our work. Text changes made to *Introduction* section (line numbers refer to the revised manuscript) are as follows:

- Lines 33-36:
  "Usually, for atmospheric radars, the surface is considered a disturbance; an important matter is to assess how large this disturbance is, i.e. to quantify the signal-to-clutter ratio. It is therefore timely to investigate and assess how beneficial such a scanning configuration could be in terms of reducing/enhancing the signal-to-clutter ratio for all type of surfaces (and not only for 'flat' oceanic)."

- Lines 42-52:
  "The mission requirements for the horizontal component of the line-of-sight wind measurements is in the order of 2.5 m/s: to achieve this goal, the contribution on mispointing errors must be lower than 0.4 m/s after all possible calibration methods (ESA-WIVERN-Team, 2023). The on-board attitude determination and control system can provide pointing and

knowledge of it within a certain degree of accuracy, which may not be sufficient to satisfy the scientific requirements. Thermoelastic deformations of the antenna will also largely contribute to the pointing error as demonstrated by the recently launched EarthCARE Doppler radar (Kollias et al., 2023). This effect is cyclical with the orbital period but it is hard to model and predict via numerical models driven by the antenna properties (e.g. by its temperature at different locations). These effects can be corrected by external calibration methods (Scarsi et al., 2024), with the surface being the simplest natural target. However, in presence of real land surfaces, clutter Doppler velocity and reflectivity profiles are expected to deviate significantly from the profiles obtained for homogeneous flat surfaces for two reasons: [...]"

- Lines 58-64:
"Better understanding the shape of the clutter reflectivity and of the clutter mean Doppler velocity profiles is paramount for two reasons: 1) the reflectivity profile can be used for geolocation purposes \citep{Pui:24} and its shape is relevant for assessing the blind zone of a radar system (i.e. the region where the radar signal will not provide any useful information for the hydrometeors); 2) the surface Doppler can be used in a data-driven approach to mitigate mispointing errors. Since the calibration of antenna distortions can occur on shorter time scales more frequent calibration points are needed. Therefore, it must be assessed what surfaces can be useful for this kind of purpose by quantifying the limits of acceptable variability in terms of sigma-zero and orography."

- Line 71-81:
"This clutter simulator represents a module of a larger end-to-end simulator endeavour being developed as part of the phase A activity funded by ESA, which simulates the full return from both atmospheric and surface targets. The whole simulator is based on the work already developed at Politecnico di Torino in the past five years (Battaglia et al., 2022b; Rizik et al., 2023; Battaglia et al., 2024). This work completes the simulator adding a thorough treatment of the surface accounting for variability of $\sigma_0$ at fine scales and orographic effects. In the previous simulator, the surface was treated in a simplistic way (flat and homogeneous) which is sufficient for oceanic surfaces. The simulator discussed in the present work will be integrated into the existing one for detailed studies that require a complete surface characterization. To the best of the authors' knowledge, there is currently no clutter simulator of reflectivity and Doppler signal for spaceborne radars taking into account NUBF for ground return and orography that has been developed for past (GPM, CloudSat, EarthCARE) or future missions (e.g. INCUS). Therefore, this type of work is a first, and it is very relevant for the WIVERN radar, given its Doppler capabilities and conical scanning operation."

Changes made to *Summary and conclusions* section (line numbers refer to the revised manuscript) are as follows:

- Lines 317-319:
"This works expands on the existing end-to-end simulator which simulates the radar observations of atmospheric and surface target by using outputs from numerical weather prediction models, expanding on the currently simple implementation of the ground clutter signal."

- Lines 330-335:
"Indeed, these aspects are of great importance for mispointing corrections of the Doppler signal. Data-driven calibrations using surface Doppler velocity measurements as an external calibration reference are the most effective, as demonstrated from ongoing work on the EarthCARE CPR. Our study has quantified the expected standard deviation of the surface $v_D$, for given NRCS and elevation variabilities and for a given integration length. Such values can be used to evaluate the use of such surfaces as reference points for Doppler calibration. This

methodology has already proven very valuable for the calibration of the recently launched EarthCARE Doppler radar."

- Lines 347-349:
  "For such systems, detailed simulations of the ground clutter and consequent refinement of clutter removal algorithms could pave the way to a better understanding of near-surface hydrometeor processes (e.g. orographic precipitation)."

**The following are other suggested corrections:**

**Lines 33-34: Introduction states "It is therefore timely to investigate and assess how beneficial such a scanning configuration could be in terms of reducing the signal-to-clutter ratio." This topic is not addressed again in the manuscript and it is not clear how the simulator in its current state would contribute to such an investigation without simulating the atmosphere. Please clarify how the clutter simulator addresses this issue. Specifically how can the simulator be used in its current state when attenuation and scattering above the surface are neglected.**

The simulator is intended as a module of a larger end-to-end simulator which aims at simulating the radar observations using results from numerical weather prediction models, expanding on the currently simple implementation of the ground clutter signal. The simulator of course accounts for attenuation and scattering in the atmosphere. For the surface clutter the only impact from the atmosphere will come from attenuation and thus to a reduction of the clutter SNR (which is usually high).

Revisions in the introduction and conclusions sections made regarding the previous comments also address this issue.

See previous response above for changes made to also address this point, in particular in lines 58-64 and 317-319 of the revised manuscript.

Furthermore, in the *Methodology* section, in lines 115-118 of the revised manuscript, the following text has been added: "No attenuation effect has been included in the clutter simulation (it is included, however, in the full simulator). The Doppler signal is not affected by attenuation as far as the SNR remains high, as is the case for surface targets, and the shape of the reflectivity profile is also unchanged (atmospheric attenuation simply lowers the profile by the path integrated attenuation of the whole atmospheric column)."

**Line 60: NRCS model is an input that should be included in the description**

Figure 2 has been updated following this recommendation. The "NRCS model LUTs" block has been added to the inputs.

In lines 88-89 of the revised manuscript, "6) a NRCS model for each surface class, based on LUTs from literature" has been added to the inputs description.

**Eqns 1, 4, 6 are stated as functions of r (LHS) but written as functions of t (RHS), please make them consistent.**

The issue has been fixed by substituting $2r/c$ in Eq. 1 with $2\xi/c$ (where $\xi$ is the distance between the infinitesimal element $dS$ and the radar, this remark has been added in line 109 of the revised manuscript), then $t$ has been changed to $2r/c$ in Eq. 1, 4 and 6.

**Line 85: Indicate what limitations the statement "No attenuation effect has been included" places on the utility of the simulator.**

Attenuation is easy to take into account, and it has been considered in the full simulator. The Doppler signal is not affected by attenuation as far as the SNR remains high, as is the case for surface targets. The shape of the reflectivity profile is also unchanged (atmospheric attenuation simply decreases the profile by the path integrated attenuation of the whole atmospheric column).

Changes made in lines 115-118 of the revised manuscript also address this point (see previous response).

**Line 89: Justify choice of flat (plane) integration implicit in del x_ij del y_ij formulation of infinitesimal and why that is an appropriate choice even though spherical earth assumption is used otherwise in the model.**

$\Delta x_{ij}$ and $\Delta y_{ij}$ are the sides of the rectangular planar elementary elements (pixels) considered in the integration, and they depend on latitude and longitude. They are calculated as $\Delta x_{ij} = R_E \cos(lat)\, \delta_{lon}$ and $\Delta y_{ij} = R_E\, \delta_{lat}$, where $R_E$ is the Earth's radius, $\delta_{lat}$ is the DEM resolution in latitude and $\delta_{lon}$ is the DEM resolution in longitude. The 3D orientation of the pixel and the stretching that originates from the orographic variability is taken into account by the term $\cos(\beta_{ij})$, which is the slope of the pixel with respect to the local tangent plane to Earth's sphere. Therefore, the area of the pixel changes taking into account the DEM variability in height, as the terms used to calculate $\beta_{ij}$ are expressed in the ECI frame. To summarize, the surface, locally at each pixel, is approximated with inclined facets whose inclination is given by $\beta_{ij}$ which is computed according to the local slope of the terrain. This approximation is deemed appropriate since the resolution of the DEM is 30 m.

In lines 121-123 of the revised manuscript the following text has been added: "The surface, locally at each pixel, is approximated with inclined plane facets of area $dS_{ij}$ whose inclination is given by $\beta_{ij}$ which is computed according to the local slope of the terrain. The area $dS_{ij}$ can be computed as a function of the DEM pixel area $\Delta x_i \Delta y_j$ by: [...]"

**Line 139: The method proposed in Battaglia et al. (2024) is cited but that reference is currently unavailable (submitted) so this method cannot be evaluated in the context of this manuscript.**

The methodology developed was already explained in *Battaglia et al., "Observation error analysis for the WInd VElocity Radar Nephoscope W-band Doppler conically scanning spaceborne radar via end-to-end simulations", Atmos. Meas. Tech, 2022*, and further improved in Battaglia et al., 2024. The paper has been accepted with minor reviews and should be out soon. A version of the paper in the accepted form can be sent to the reviewers in confidential form.

The first reference has been added to the bibliography and the text changed in line 174-175 of the revised manuscript as follows: "[...] real Doppler and reflectivity signals are generated according the method proposed by Battaglia et al. (2022b) and further improved in Battaglia et al. (2024)."

**Line 157-159: The choice of these two cases is important in understanding the applicability of the work. The results would be strengthened by citation or further justification for the choice of the**

**high correlation case. This is needed to give context to the conclusion "high correlation value … produces much better results and seems very promising" (line 231-232). rho on these lines should be changed to rho_HV**

The two cases result from two different methods to calculate the Doppler signal return. The first originates by correlating the H and V signals from the actual surface range. Unfortunately for land surfaces the correlation between the H and V reflection signals is low (of the order of 0.5 as observed from airborne observations, Wolde et al., 2019). The second originates from correlating the surface ghosts (Rizik et al., 2023), which have a much higher correlation but lower signal-to-noise ratio (but still much larger than 0 dB) because of the large sigma-zero of land surfaces. The figure below better explains these two different methods to derive the estimates of the surface Doppler.

[Figure]

*Figure 1: schematic describing the two possible methods for deriving surface Doppler based on pulse pair estimates (the black lines indicate the signals that are correlated).*

In the first case co-polar signals at the same range (the surface in figure, at $r_{surf}$) in the H and V channels are used: this offers higher SNR compared to the other method (by 5 dB, 10 at maximum) but the correlation is lower (around 0.5) due to the scattering characteristics of land surfaces. In the other case, the two cross-talk signals are used at different ranges, in particular separated by a range equal to $2\Delta r_{T_{HV}} = 4T_{HV} \, c$, where $T_{HV}$ is the interval between H and V pulses, 20 $\mu s$ (this value was missing from Table 1, it has been added in the revised manuscript). The two co-polar signals at the surface range correspond to the H (first) and V (second) separate pulses sent $T_{HV}$ one after the other bouncing back from the surface without changing polarization. The two cross-talk signals originate from the same H or V pulses which are backscattered in the cross polarization and therefore appearing at different ranges (higher above or below the surface). For these signals the return power is lower but the correlation is much higher because of reciprocity.

In the revised manuscript, Section 2.3 has been changed to better address this aspect, with the addition of the figure above (as Figure 6 in the revised manuscript) and of the following text in lines 197-201 of the revised manuscript:

"These two cases are better illustrated in Fig. 6. The two co-polar signals at the surface range correspond to the H (first) and V (second) separate pulses sent $T_{HV}$ apart, reflecting back from the surface without changing polarization. The two cross-talk signals originate from the same H or V pulses which are backscattered in the cross polarization and therefore appearing at different ranges (higher

above or below the surface). For these signals the return power is lower but the correlation is much higher because of electromagnetic reciprocity."

The mistake about the $\rho_{HV}$ notation has been fixed in the revised manuscript in lines 192 and 194.

**The case studies section should be reworked with emphasis given to readability.**

The case studies section has been reworked in the revised manuscript.

Various text changes have been made in lines 204-259 (sections 3.1 and 3.2) of the revised manuscript, see the marked-up version of the manuscript for the exact changes.

**Line 242: Please describe the renormalizing procedure.**

The height at which the boresight intercepts the surface of the DEM (i.e. at the range of the surface along the boresight) is subtracted from each profile height.

In lines 297-298, the following phrase has been added to point out the procedure: "[...], by subtracting from the height of each profile the height at which the boresight axis intercepts the DEM surface."

**Line 274-276: Indicate what value would be gained by improving the NRCS dataset, specifically what benefit would justify an additional field campaign. Also indicate what value the simulator would bring to the EarthCARE and CloudSat missions.**

The NCRS dataset used to build the LUTs of Fig. 4 is outdated, coming from experimental campaigns carried out in the 1980s. At that time there was not much interest in higher frequency bands. Higher incidence angles are missing (the sampled incidence angles are also sparse) and for some class terrains the Ka or Ku band values had to be used due to missing data in the W band. In particular it is very important to better establish the drop in NRCS when moving from nadir to very slant angles on surfaces like different types of snow, sea ice and different land biomes.

The Doppler value from the surface can be used as a reference for calibration of mispointing of the antenna in real applications, such as the EarthCARE mission. Also better understanding the clutter impact on the hydrometeor profiling can be of interest, e.g. for orographic precipitation studies.

These observations have been added to the revised manuscript in the *Conclusions* section in lines 340-344 with the following text:

"The NCRS dataset used to build the LUTs in Fig. 4 is based on experimental campaign carried out in the 1980s. Higher incidence angles are missing (the sampled incidence angles are also sparse) and for some class terrains the Ka or Ku band values had to be used, due to missing data in the W band. For systems adopting high incidence angles like WIVERN it will be critical to better establish the drop in NRCS when moving from nadir to very slant angles on surfaces covered with different types of snow, sea ice or different land biomes."

**Reply to Anonymous Referee #2**

*We thank the anonymous referee for his review and his detailed comments.*

*Below, the referee's comments can be found in bold, followed by the authors' replies.*

**Simulators are useful for predicting the performance of instruments and helpful in understanding various error sources and in devising algorithms to extract the maximum amount of information. In this paper a simulator for the proposed WIVERN W-band radar is used to examine the behavior of the reflectivity and Doppler profiles over mountainous terrain.**

**Although I have several questions on the details, I found the paper informative. Since the WIVERN radar will be used primarily for cloud sensing, I expected to see some results on the atmospheric effects on surface cross section and Doppler but perhaps that will be dealt with in a separate paper. I recommend publication after the authors address the comments below.**

**Table 1: It's not clear to me whether the radar will transmit H and receive H and V (and transmit V and receive H and V) or whether it will transmit H and receive H only and transmit V and receive V only.**

The radar will transmit and receive both H and V signals in pairs. Each pair has a repetition frequency of 4 KHz, with a delay between the H and V pulses of 20 microseconds. This value was missing and has been added to Table 1 as $T_{HV}$ in the revised manuscript. Figure 1 of Rizik et al. (2023), shown below, offers a good representation of the pulse sequence for the WIVERN polarization diversity radar. A reference to this figure has been added to the revised manuscript.

[Figure]

*Figure 2: Two pulses with different linear orthogonal polarizations are sent with a short pulse-pair interval, T_HV. The dashed line corresponds to the cross-talk of the leading pulse which interferes with the trailing pulse. The shaded regions correspond to range*

In line 97-98 of the revised manuscript, the following text has been added:

"The radar will transmit and receive both H and V signals in pairs. Each pair has a repetition frequency of 4 KHz, with a delay between the H and V pulses of 20 $\mu s$ (see Fig. 1 in Rizik et al. (2023) for a schematic of the concept)."

**More generally, I'm not clear about the meaning of the 'ghost pulse' and how this affects the Doppler processing. This issue comes up later in the paper where the rho(HV) parameter, which implies that the cross-pol will be measured, is used to categorize the results. A few more sentences would be**

**helpful to explain how this parameter is related to the scattering properties of the surface.**

Ghost pulses originating from scattering of H or V pulses in the other channel can be used to calculate the Doppler, using the cross-talk signals. This sacrifices SNR in the return power to obtain a less noisy Doppler signal.

The issue has been explored in depth in a reply to Anonymous Referee #1 (see Figure 1 in this document and associated discussion), with an explanation and a graphical representation of the process and the cases adopted for computation of the Doppler signal.

In the revised manuscript, Section 2.3 has been reworked to better address this aspect. A figure has also been added (Figure 1 in this document).

**Fig. 4. The DPR (dual-frequency precipitation radar on the GPM satellite) surface cross-section data (Ku/Ka-band) over land shows a sharper decrease with angle in moving off nadir than the results shown here. The DPR data covers the angle range from nadir to 18 deg so airborne data are needed to fill in at the higher incidence angles.**

**I couldn't remember how sharp the drop-off with angle was but found the figure below. The land data are not categorized by surface type – all surface types are included.**

**The radar frequency is not mentioned in Fig. 4 and while the DPR database shows the Ku & Ka-band are similar in their angle dependencies over both land and ocean, the W-band data might depart significantly from the lower frequency data.**

**Unfortunately, there doesn't seem to be much off-nadir sigma-zero data at W-band, at least that I'm aware of. One advantage of the simulator is that the surface scattering model can be updated as new information becomes available.**

[Figure]

**Mean values of DPR sigma-zero from 1-month of data are shown on the left for ocean (top) and land**

(bottom).  These data were taken from 35° S to 35° N to match the TRMM coverage.  The Ka-band cross section over land decreases by about 12 dB going from nadir to 9°.  (Since the data used here were measured early in the mission, the Ka-band data extended only to 9°.)

The data adopted to develop the look-up tables comes from experimental campaign in the 1980s. For consistency, the data was taken from the same source, which included extensive tables for different types of surfaces and frequencies. Unfortunately, data for W band was not available for all classes, so data for Ka or Ku band was adopted for some of them as a best approximation.

We are aware of the GPM measurements. Apart from the routine measurements *K. Yamamoto et al., "A Feasibility Study on Wide Swath Observation by Spaceborne Precipitation Radar," IEEE Journal of Selected Topics in Applied Earth Observations and Remote Sensing, 2020* describes results for a wide swath experiment with GPM performed in Ka band, confirming the drop of sigma-zero up to +/-18 deg, for large footprints (in the order of 5 km). The experiment extended the sigma-zero characterization up to 35 deg scan angles. For the ocean the drop is consistent and for land there seems to be a slight decrease too. However, land classification is not present in the above-mentioned paper, and the data may be unreliable due to the fact that the GPM electronically scanning antenna was not designed to look at those scan angles (up to 35 deg).

In general, given WIVERN scanning geometry, better coverage at around 40 deg incidence is crucial. We are aware of the limitations in our NRCS model, and it is indeed true that it can be easily updated with newer data as soon as it is available.

**Lines 148-155.  I had trouble following this discussion.  First, it would be clearer to say something like: 'Land surfaces are generally categorized by large values of depolarization (-10 to -3 dB) and low values of rho(HV) (0.4 to 0.8).'**

We agree, lines 184-185 in the revised manuscript have been modified following this recommendation.

**My confusion comes in the next sentence. 'While there is not much correlation for the co-polar surface signals ..'** *(does this mean between rho(HH) and rho(VV))* **'...there is an excellent correlation between the cross-polar signals ...'. – italics mine. But the previous sentence stated that these values are low so I must be missing something.**

See Figure 1 of this document and associated discussion.

**Is 'rho' in lines 157 and 159 the same as 'rho(HV)' in the previous paragraph?  If so, the same notation should be used.**

Correct, the notation has been fixed in the revised manuscript.

**Fig. 6 caption: 'wo'→'two'.**

The mistake has been fixed in the revised manuscript.

**Line 166: I see a blue 'X' but not a black cross.**

The mistake has been fixed in the revised manuscript.

**I'm a bit confused by Fig. 7 and the associated discussion. It is assumed that the antenna is pointed in a direction orthogonal to the satellite velocity vector so if the sigma-zero were uniform over the footprint, then the Doppler would be zero - as shown by the red line. The variation in range in the Doppler is presumably caused by NUBF so a positive Doppler (assuming positive is toward the radar) would be caused by the return power from the forward portion of the beam being larger than that from the backward portion of the beam. Is this correct?**

Yes, it is correct. Return power is directly related to the $w_{ij}^{v}$ quantities plotted in fig. 8, as shown in Eq. (6). This equation fundamentally says that the Doppler velocity at a given range is the result of the weighted average for any given surface domain corresponding to a given range of the satellite velocity projection along the line-of-sight (value in the red iso-lines in Fig. 8) with the weights $w_{ij}^{v}$ (color coded in Fig. 8). These weights indicate the return power reflected to the radar by a given surface pixel, which results from combination of NRCS and the square of the antenna gain.

A revision has been made to better illustrate this aspect pointed out in the comment. In line 219-222 of the revised manuscript the following text has been added:

"The principle is better outlined considering Eq. (6), which highlights that the Doppler velocity at a given range is the result of the weighted average, for all surface domains corresponding to that given range, of the satellite velocity projection along the line-of-sight (value in the red iso-lines in Fig. 9 with the weights $w_{ij}$. The weights correspond to the return power reflected to the radar by a given surface pixel, which results from a combination of NRCS and the square of the antenna gain. "

**I would have expected the reflectivity profile to be much more variable in range than the blue line shown in Fig. 7 left panel. How typical is this; how much does it change when a field of view over the mountains is taken?**

In this case, the radar illumination comes from the right of the scene. Although in the radar footprint there is a huge inhomogeneity in NRCS, this is not reflected in a strong difference between the reflectivity profile (blue line in Fig. 8) and the reflectivity profile of a flat homogenous surface (red dashed line in Fig. 7) because the iso-range lines are almost parallel to the gradient of sigma-zero. A profile much more variable in range can be obtained for this scene by changing the illumination geometry, e.g. aligning the boresight with the sigma-zero gradient. Otherwise, this scene is rather flat in terms of orography, compared to much more mountainous parts of the Piedmont regions, e.g. the Alps. For the latter conditions, it is also possible to find case studies where vertical profile of reflectivity strongly departs from the flat homogenous refence shape (see Fig. 9, top right panel, in the orographic segments at the beginning and the end of the scan).

An idea of how the reflectivity profiles are deformed in mountainous regions is given by the two top panels of Fig. 11 (though these are CFADs). On the left the profiles are close to the flat homogenous case, while on the right cases with high elevation standard deviation are considered.

These observations have been incorporated into the revised manuscript with the following additions (line numbers refer to revised manuscript):

- Lines 231-235:
  "An additional observation can be made: in Fig. 8 the real reflectivity profile in blue does not show much variation with respect to the red dashed line (although some change in the power distribution can nevertheless be seen). This can be explained by looking at Fig. 9 and the right panel of Fig. 7: the $\sigma_0$ gradient is roughly parallel to the iso-range lines, so NUBF does not affect

the reflectivity integration inside each annulus. More variability would have been obtained if the same scene were looked at in a forward or backward view at an azimuth of 0° or 180°"

- Lines 257-259:
"In orographic regions at the beginning and end of the scan, the reflectivity profiles deviate notably from the flat homogeneous shape, unlike the profiles in the middle segment, originating from a flatter portion of terrain."

**Not sure if side lobes are included in the antenna pattern but these would add to the Z variability, especially in the mountains.**

Yes, sidelobes are considered in the computation of the return power, up to -30 dB (normalized gain). Fig. 8 and the right panel of Fig. 6 show gain contours in black, with the presence of the first sidelobes which is 0.17 deg from the boresight peaking to a normalized gain of -24 dB.

As can be seen by the weight magnitudes in Figure 8, the contributions of the region illuminated by the antenna sidelobes are much smaller than those from the region illuminated by the main lobe.

A note has been added to the revised manuscript at line 208 to remark this aspect: "Sidelobes contributions are included up to -30 dB (see Fig 7 and 9)."

**Fig 9 caption on explanation of the bottom two panels, right-hand side. Presumably, means are given by the blue lines and std dev's are given by the red lines. This should be mentioned.**

In the revised manuscript this information has been pointed out in the caption as recommended.

**Does '1 km averaging region of elevation' mean that for calculation of the surface cross section the radar return power is used over a 1 km range window to compute mean and std dev? For example, if the mean & std dev at a particular point are (15, 5) dB, does this mean that about 66% of the data falls within 10 to 20 dB?**

For computation of the mean and standard deviation values, a square grid of 1km×1km centered on the point hit by the boresight axis on the DEM surface is used. This was done in order to account for the region that contributes more to the computation of the Z and $v_D$ profiles (outside from this region the antenna gain is very small). Using this window, which is running along the boresight track, the mean and standard deviation of sigma-zero in dB and elevation in metres is obtained computed and attributed to the central pixel of the window domain.

**Does the phrase 'with the antenna scanning at the side of the satellite ground track' mean that the data are taken at an azimuthal angle at 270 deg?**

The azimuthal angle is around 90 deg, and the satellite is moving northbound. We agree that the phrase is rather confusing, in the revised manuscript it has been substituted with "in side configuration, with a scanning azimuthal angle of about 90 deg".

**Line 209: correspondence**

The mistake has been fixed in the revised manuscript.

**line 213: '.. this value is expected to be zero (*for fields of view orthogonal to the direction of spacecraft motion*)..' . - italics mine. Although this was noted earlier, I think it's important to emphasize that the direction along the incidence angle is perpendicular to the spacecraft motion.**

Regardless of the direction of the boresight during its rotation, once the satellite orbital velocity projected along the boresight is subtracted from the Doppler profile, the value corresponding to the surface range along the boresight (assumed to be at the peak of reflectivity) is expected to always be zero for flat and homogenous terrain. Departures from the zero value can be attributed to noise and NUBF.

**Line 218: should the fourth category be: 7 dB < std(s0) < 25 dB?**

Correct, the mistake has been fixed in the revised manuscript.

**Use of 'dB' here and use of 'meter' for the std dev of height might make this more readable and remind the reader of the units.**

Yes, this recommendation has been followed in the revised manuscript. In lines 271-273 of the revised manuscript the units have been added.

**Line 220: 'A few' rather than 'Few'. 'Few' implies 'Only a few'.**

Corrected in the revised manuscript.

**Lines 221-222: point 1 is difficult to understand and should be rewritten.**

Point 1 has been rewritten in the revised manuscript in lines 275-277: "The classes have been chosen to include a significant number of occurrences ($N$ inside the boxes), but, as the terrain in the chosen region is relatively flat, in general classes with smaller standard deviation in elevation present more occurrences."

**One possibility: The classes have been defined to include a significant number of cases in each. Those classes where the standard deviation in elevation is small have a high-count number because much of the terrain in the segment chosen is relatively flat.**

Yes, this is correct. Even though the standard deviation classes have been spaced so a reasonable number of occurrences are present for each class, the ones at lower elevation std have a higher count due to the region chosen as case study being predominantly flat.

**(The unwritten assumptions are that low std dev in elevation implies relatively flat terrain which implies a small standard deviation in sigma-zero. But I think these assumptions are OK.)**

Orography does impact the variability of $\sigma_0$ due to the variability in incidence angle, but $\sigma_0$ variability is mainly impacted by the terrain class diversity. In Section 3.1 it is demonstrated that even slight orography induces NUBF and therefore departure from 0 m/s due to changes in range and antenna gain.

**Fig. 11: I'm having trouble understanding the behavior of the Doppler in the middle figures. From the title of the left middle figure, it seems that phi(A) is being varied from -15 to 15 deg but wouldn't the Doppler be the same regardless of a change of sign in phi(A)? What parameter is being changed to produce the positive and negative Doppler.**

The Doppler profiles at around 0 deg and 180 deg (forward and backward, respectively) in theory have the same shape (approximately), but the sign of the slope is opposite. The Doppler profiles with $\phi_A$ in the [165, 185] deg range have been inverted in sign in order to group them together as done with the two side configurations (two bottom panels).

**For the forward-looking case, the large Doppler shift induced by the satellite motion has been subtracted off, correct?**

Yes, correct.

**Line 242: renormalised**

The mistake has been fixed in the revised manuscript.